

# Extending and understanding the South West Western Australian rainfall record using a snowfall reconstruction from Law Dome, East Antarctica

Yaowen Zheng[1,2], Lenneke M. Jong[3,4], Steven J. Phipps[1], Jason L. Roberts[3,4], Andrew D. Moy[3,4], Mark A. J. Curran[3,4], and Tas D. van Ommen[3,4]

[1]Institute for Marine and Antarctic Studies, University of Tasmania, Hobart, Tasmania, Australia
[2]College of Oceanic and Atmospheric Sciences, Ocean University of China, Qingdao, Shandong, China
[3]Australian Antarctic Division, Kingston, Tasmania, Australia
[4]Australian Antarctic Program Partnership, Institute of Marine and Antarctic Studies, University of Tasmania, Hobart, Tasmania, Australia

**Correspondence:** Yaowen Zheng (yaowenz@utas.edu.au)

**Abstract.**

South West Western Australia (SWWA) has experienced a prolonged reduction in rainfall in recent decades, with associated reductions in regional water supply and residential and agricultural impacts. The cause of the reduction has been widely considered, but remains unclear. The relatively short length of the instrumental record limits long-term investigation. A previous

proxy-based study used a statistically negative correlation between SWWA rainfall and snowfall from the Dome Summit South (DSS) ice core drilling site, Law Dome, East Antarctica and concluded that the anomaly of recent decades is unprecedented over the ∼750 year period of the study (1250–2004 CE). Here we extend the snow accumulation record to cover the period 22 BCE–2015 CE and derive a rainfall reconstruction over this extended period. This extended record confirms that the recent anomaly is unique in the period since 1250 CE and unusual over the full ∼2000 year period, with just two other earlier droughts

of similar duration and intensity. The reconstruction shows that SWWA rainfall started to reduce around 1971 CE. Ensembles of climate model simulations are used to investigate the potential roles of natural variability and external climate drivers in explaining changes in SWWA rainfall. We find that anthropogenic greenhouse gases are likely to have contributed towards the SWWA rainfall drying trend after 1971 CE. However, natural variability may also have played a role in determining the timing and magnitude of the reduction in rainfall.

## 1 Introduction

South West Western Australia (SWWA) is characterized as a region with a Mediterranean-like climate (Yu and Neil, 1993; Siddique et al., 1999; Timbal et al., 2006). Most of the rainfall occurs during the winter and spring seasons, comprising around 70% of the annual total rainfall (Wright, 1974a; Hennessy et al., 1999; Ludwig and Asseng, 2006; Timbal et al., 2006). The seasonal concentration of rainfall makes May to October (approximately) the growing season for SWWA (Watson

and Lapins, 1969; French and Schultz, 1984; Anderson et al., 1996; Ward and Dunin, 2001). In comparison to other regions of



Western Australia (WA), SWWA has a more constant supply of water throughout the year for agriculture, industry and residents (Wright, 1974a, b; Anderson et al., 1996; Pitman et al., 2004; Power et al., 2005; Ludwig and Asseng, 2006). Abundant rainfall contributes to suitable conditions for dryland farming such as wheat with a yearly sowed area of more than 4,000,000 ha (Ludwig and Asseng, 2006). It also directly provides water supply to the WA capital city Perth (Pitman et al., 2004; Power

et al., 2005). A number of studies have found the growing season rainfall in SWWA has decreased (Pittock, 1983; Li et al., 2005; Cai and Cowan, 2006; Samuel et al., 2006; Timbal et al., 2006; Ludwig et al., 2009; Feng et al., 2010). Rainfall in this region has decreased by 22% in August and 30% in September for the period 1946–1978 compared to the period 1913–1945 (Pittock, 1983), and has weakened by about 20% from the 1960s to the beginning of the 21st century (Cai and Cowan, 2006). Winter rainfall has decreased by 15%-20% from the 1970s to the early 21st century (van Ommen and Morgan, 2010). This

sizable winter rainfall reduction contributes to decades of persistent drought since 1975 (Ansell et al., 2000; Cai and Cowan, 2006; Hope et al., 2006) causing water supply problems in WA (Pitman et al., 2004; Samuel et al., 2006). The SWWA rainfall reduction occuring in the mid-20th century decreased the water supply to Perth by about 42% (Pitman et al., 2004). This has required an additional investment of around $300 million to develop alternative water sources (Pitman et al., 2004). This long-term ongoing drought poses a potential threat to residential water supplies, industrial and agricultural production and makes

the study of this phenomenon and the determination of its driving factors urgent.

The large-scale climate drivers for both Australia and specifically the SWWA region have been investigated over the past decades. The El Niño–Southern Oscillation (ENSO), specifically the equatorial Pacific Ocean sea surface temperature anomaly in the Nino 3.4 region (Trenberth and Hoar, 1997), tends to be associated with interannual rainfall variations in Australia (Cai et al., 2011), including dry conditions (Chiew et al., 1998). However, this teleconnection linking ENSO and Australian rainfall

is stronger in Eastern Australia than in SWWA. The Indian Ocean Dipole (IOD) shows a robust relationship during the growing season from May to October (Fierro and Leslie, 2013) with the rainfall in southern and western regions of Australia (Ashok et al., 2003), but not specifically to the SWWA region. The Southern Annular Mode (SAM), or the Antarctic Oscillation (AAO), is a large-scale mode of climate variability that is correlated with rainfall in WA (Gong and Wang, 1999; Thompson and Solomon, 2002; Fierro and Leslie, 2013). The relationship between the SAM and SWWA rainfall is seasonal, with the

SAM influencing rainfall in June-July-August (JJA) but not in December-January-February (DJF) (Cai and Shi, 2005; Cai and Cowan, 2006). The SAM has experienced a shift towards a more positive phase since the 1970s, which can be attributed to the depletion of stratospheric ozone (Thompson and Solomon, 2002; Gillett and Thompson, 2003). This shift, in conjunction with the increase in anthropogenic greenhouse gases over this period, may be responsible for at least part of the reduction in SWWA rainfall (Cai and Shi, 2005). However, changes of a similar magnitude to those observed can potentially also arise through

natural multidecadal climate variability (Cai and Shi, 2005). Thus the drivers of the winter rainfall attenuation in SWWA are still unclear.

The relatively short length of the observational record in SWWA limits long-term studies of SWWA rainfall. Most of the rainfall-related studies undertaken to investigate the SWWA region used data from the Australian Government Bureau of Meteorology (BoM). The original BoM rainfall data in SWWA is the instrumental rainfall station record. The earliest record

goes back to around the 1880s, while most of the station records started at around 1900. The availability of 120 years of rainfall





data is insufficient to investigate the SWWA long-term rainfall evolution in history and makes it difficult to determine the uncertainties of the climate drivers in SWWA. A relationship between rainfall in SWWA and the snowfall recorded in Dome Summit South (DSS) ice core drilling site on Law Dome, East Antarctica was found by van Ommen and Morgan (2010). Additionally, the strength and position of southern hemisphere westerlies dominate changes in coastal Antarctica snowfall, so

we may expect an inverse relationship between SWWA rainfall and coastal Antarctica snowfall rates, especially for cyclonically driven locations such as Law Dome, East Antarctica (Bromwich, 1988). The relationship consists of a statistically significant negative correlation between winter JJA mean SWWA regional rainfall and Law Dome, East Antarctica (van Ommen and Morgan, 2010). The robustness of the correlation was strengthened by 5-year smoothing (van Ommen and Morgan, 2010). More recently, a 2035-year long-term ice core annual record of snow accumulation rates for Law Dome has been published

(Roberts et al., 2015). This record was dated by determining annual layers in the seasonally varying water stable isotope ratios and trace ions from multiple ice cores drilled at the DSS site (66.7697°S, 112.8069°E, 1370 m elevation) (Roberts et al., 2015). Taken together, the relationship between SWWA rainfall and DSS snowfall and the DSS long-term ice core record allows us to extend the SWWA rainfall record to span the past 2000 years.

        In this study, we mainly focus on reconstructing growing season (May to October) SWWA rainfall over the past two thousand

years and answering the questions about changes in SWWA rainfall since around 1970s and the drivers of these changes by comparing observational data with climate model simulations. We calculate and test the significance of the correlation between growing season SWWA rainfall and DSS snow accumulation. We individually calculate and test the significance of the correlation based on 116 years of the Australian Water Availability Project (AWAP) gridded growing season rainfall data for 110 Local Government Areas (Australian Government, 2020) and combine the statistically significant ($p < 0.05$) areas to

define the SWWA region of significance. We assume stationarity and build a linear model between the growing season rainfall in the significant region and the DSS snow accumulation and use this model to reconstruct the growing season rainfall in SWWA from 2015 CE back to 22 BCE. We then compare the reconstruction with ensembles of simulations conducted using the Commonwealth Scientific and Industrial Research Organisation Mark 3L (CSIRO Mk3L) model (Phipps et al., 2011, 2012, 2013). This allows us to explore the drivers of the changes in rainfall in SWWA rainfall and place them in a longer-term context.

## 2    Data

### 2.1    DSS ice core record

DSS annual snow accumulation is calculated by dating the ice cores drilled at DSS (Roberts et al., 2015), East Antarctica. The length of this DSS snow accumulation record has been extended to 2035-years, from 2012 CE back to 22 BCE (Roberts et al., 2015).

In addition, the latest DSS snow accumulation data has been extended to 2015 CE using a recent new ice core (DSS1617). We use the annual layer thickness data from this new core (DSS1617) for the period 1990 to 2015 CE to extend and replace the DSS1213 annual layer thickness data for the period 1990—2012 used in Roberts et al. (2015), and calculate the corresponding



**Table 1.** Observational data types include two DSS snow accumulation records, AWAP gridded rainfall and BoM stations data.

| Data type | Time period | Reference |
|---|---|---|
| Law Dome Summit accumulation composite⋆ | 22 BCE–2012 CE | Roberts et al. (2015) |
| Law Dome Summit accumulation composite∧ | 22 BCE–2015 CE | This study |
| AWAP Gridded rainfall | 1900–2018 CE | Jones et al. (2009) |
| BoM station data | around 1900s–2019 CE | Lavery et al. (1997) |

⋆DSS accumulation composite – DSS-main, DSS99, DSS97 and DSS1213.

∧DSS accumulation composite – DSS-main, DSS99, DSS97 and DSS1617.

snow accumulation record using their Power-law vertical strain rate model. Therefore, the DSS snow accumulation record we use in this study will be a 2038-year long-term record from 22 BCE to 2015 CE.

## 2.2 SWWA instrumental rainfall record

The SWWA rainfall station records are monthly records which are obtained from BoM. We initially selected 16 stations, which are as the same as van Ommen and Morgan (2010). Since the instrumental rainfall records for station Avondale Farm (ID: 10795. 116.87°E 32.12°S) are only available for 51 years (1965–2015), which is substaintially less than the average of remaining 15 stations records length (around 118 years), we will not discuss the rainfall records of this station any further. The other 15 stations cover the 1900s to 2019 CE.

To investigate more completely the spatial variability and reconstruct SWWA rainfall, we use AWAP Australian Gridded Climate Data (AGCD) (Jones et al., 2009). This AWAP data is gridded data that has 0.05° longitude * 0.05° latitude geospatial resolution and 119 years' availability from 1900 CE to 2018 CE.

## 2.3 CSIRO Mk3L climate model simulations

The model outputs used for paleoclimate data-model comparison are from the CSIRO Mk3L climate model. CSIRO Mk3L is a reduced-resolution coupled general circulation model, comprising components that dscribe the atmosphere, ocean, sea ice and land surface (Phipps et al., 2011, 2012). The model is explicitly designed for studying climate variability and change on millennial time scales. The atmospheric component of CSIRO Mk3L is taken from the CSIRO Mk3 climate model (Gordon et al., 2002), but with reduced horizontal resolution. Both CSIRO Mk3 and CSIRO Mk3L produce credible simulations of large-scale precipitation, including over Australia (Cai et al., 2003; Cai and Shi, 2005; Phipps et al., 2011). The model is also skilful at capturing the dominant modes of large-scale variability in the Southern Hemisphere —— ENSO and SAM —— including the teleconnections between these modes and precipitation over Australia (Cai et al., 2003; Phipps et al., 2011, 2013; Abram et al., 2014; Barr et al., 2019).

In this study, we use four ensembles of simulations based on a combination of orbital forcing (O), greenhouse gases (G), solar irradiance (S) and volcanic aerosols (V) (Phipps et al., 2013). The four ensembles (O, OG, OGS and OGSV) are generated by





combining each of these forcings (Table 2). Each ensemble contains three members, which differ only in that they are initialised from different years of a pre-industrial control simulation. As the initial states therefore differ, the internal variability will be different between each member but the responses to external forcings will be consistent.

**Table 2.** The forcing(s) and duration for each ensemble of the CSIRO Mk3L model outputs

| Ensemble | Forcing(s) | Duration |
|----------|------------|----------|
| O | Orbital | 1 CE–2000 CE |
| OG | Orbital, greenhouse gases | 1 CE–2000 CE |
| OGS | Orbital, greenhouse gases, solar irradiance | 1 CE–2000 CE |
| OGSV | Orbital, greenhouse gases, solar irradiance, volcanic aerosols | 501 CE–2000 CE |

To compare the CSIRO Mk3L simulations with the SWWA rainfall reconstruction, we use all 12 simulations, three for each of the four ensembles. The simulations have a geospatial resolution of 5.625° longitude by 3.1857° latitude. The model has been run globally for 2000 years from 1 CE to 2000 CE and the outputs have been published (Phipps et al., 2013). For 9 simulations, three members for each of the O, OG and OGS ensembles, 2000 years of output is availabile covering the period from 1 to 2000 CE. For the three members of ensemble OGSV, 1500 years of data is available covering the period from 501 to 2000 CE. The model outputs consist of monthly mean data.

All the data we use in this study, including the stations rainfall, gridded rainfall, DSS record and model outputs, are for the growing season from May to October.

## 3 Methods

### 3.1 Normality testing

We use one-sample Kolmogorov-Smirnov tests (hereafter KS test) to assess the normality of the observational data. There is statistically significant evidence that the 15 BoM stations data (interpreted periods up to 2015 CE), AWAP gridded data and DSS snow accumulation data (interpreted the period 1900 CE to 2015 CE) all fit the normal distribution corresponding to their mean and standard deviation, respectively. A visual validation, which consists of comparing the differences between the empirical and normal cumulative distribution functions (CDFs), is presented in Appendix A. The 15 BoM stations, AWAP gridded rainfall and DSS snow accumulation data empirical CDFs approximately fit their corresponded normal CDFs with minimal biases (Appendix A). With no irregularity between the empirical and normal CDFs, the visual validation for the observational data is consistent with the results of a one-sample KS test. Therefore, we are confident that the data is described by a normal distribution.



## 3.2 Significance testing and region definition

Low-pass filtering (or smoothing) the data increases the correlation of the precipitation time-series between SWWA and
Law Dome (van Ommen and Morgan, 2010), but introduces autocorrelation which reduces the temporal degrees of freedom
(Bretherton et al., 1999). In order to determine the optimal window size for running average smoothing, we test smoothing using windows sizes of 1–10 years on SWWA and DSS data and then calculate the Pearson correlation coefficients. As
smoothing reduces the temporal degrees of freedom, we estimate the Effecive Sample Sizes (ESS) or temporal degrees of
freedom (Bretherton et al., 1999) by first calculating the lag-1 autocorrelation for each sample and the apply the following
formula,

$$\text{ESS} = N \frac{1 - r_1 r_2}{1 + r_1 r_2} \tag{1}$$

where N is the sample sizes, $r_1$ and $r_2$ are the lag-1 autocorrelation coefficients corresponded to the two samples (Bretherton et al., 1999). Next, we calculate Student's t-statistic using:

$$t_S = r \sqrt{\frac{\text{ESS} - 2}{1 - r^2}} \tag{2}$$

where $r$ is the Pearson correlation coefficient and then compute the corresponding p-value. We repeat the above calculations
for both the gridded and stations rainfall data matching with DSS snow accumulation data. A six-year window (details in
Appendix B) maximizes the area of the correlation between AWAP gridded rainfall and DSS snow accumulation with statistical
significance. For the 15 BoM stations data, the number of the stations to show a statistically significant correlation is maximized
by a window size of 5-years (see Appendix B). The rainfall changes and its drivers we are interested in this study is for the
SWWA region instead of separate stations. For consistency with the AWAP gridded data, we also use a six-year window for
the BoM stations data

To define a region with statistically significant correlation between observed (AWAP) and reconstructed (DSS) rainfall
over SWWA, we independently calculate the correlation coefficient and test its significance for 110 Local Government Areas
(Australian Goverment, 2020) (details in Appendix C) in WA. There are 9 Local Government Areas smaller than the 0.05°
longitude * 0.05° latitude geospatial resolution of the AWAP data grid. Therefore, we actually calculate and test 109 areas.
There are 52 areas that are statistically significant (6-year window, p < 0.05). We combine these 52 areas to define the significant
region (Figure 1) over SWWA. For convenience, we name this significant region as "MASK".

The correlation map (Figure 1) shows the correlation coefficients for 6-year smoothed AWAP rainfall and DSS snow accumulation. The map shows that the strongly negatively correlated ($r \leq -0.5$) areas are mainly concentrated in the southwest
corner. Areas around the top right show positive correlations with no statistical significance (p > 0.05). MASK covers almost
the whole SWWA region apart from some coastal areas in the south. Southern coastal areas show no statistical significance (p
> 0.05), consistent with the findings of van Ommen and Morgan (2010) who reported the south coast local rainfall related to
onshore northward flow causing the no significant correlation with DSS.





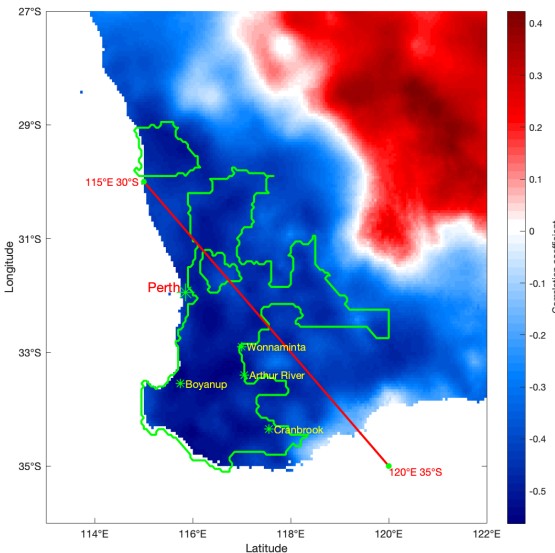

**Figure 1.** The correlation map for the southwest part of WA region for 6-year window AWAP rainfall and DSS snow accumulation from 1900 CE to 2015 CE. The outline area (green line) is the MASK region where the correlation is statistically significant ($p < 0.05$). Red diagonal line connecting 115°E 30°S and 120°E 35°S is the boundary of SWWA (van Ommen and Morgan, 2010). Perth is the capital city of WA. Boyanup, Wonnaminta, Arthur River and Cranbrook are the four significant (6-year window, $p < 0.05$) stations.

165    In order to quantify the MASK correlation coefficient and evaluate the statistical significance, we multiply the mask matrix (in the region has a value of 1 and outside the region has a value of 0) of the MASK with the AWAP gridded data to generate the MASK rainfall. Then we calculate the MASK rainfall correlation coefficient with DSS and test its statistical significance.

**Table 3.** The Pearson correlation coefficients for the MASK rainfall and the four BoM stations rainfall with the DSS snow accumulation. All the correlations are statistically significant (6-year window, $p < 0.05$).

| Sample | Correlation coefficient | Year (CE) |
|---|---|---|
| MASK | -0.597 | 1900–2015 |
| Arthur River | -0.548 | 1891–2015 |
| Boyanup | -0.623 | 1898–2013 |
| Cranbrook | -0.540 | 1891–2015 |
| Wonnaminta | -0.546 | 1905–2015 |





Table 3 integrates the results of the significance testing and correlations. There is statistical significance ($p < 0.05$) that the MASK rainfall and DSS snow accumulation are strongly negative correlated. BoM stations rainfall also show consistent results. From the 15 stations tested, there are statistically significant ($p < 0.05$) correlations with DSS for four stations. These
170    four stations are all geographically located in the MASK region (Figure 1) showing the consistency with the significance of MASK. Therefore, we are confident to construct a linear model for MASK rainfall and DSS snow accumulation.

### 3.3   Linear model construction

The scatter plot for MASK rainfall and DSS snow accumulation is shown in Figure 2a. The data show a generally linear distribution with a negative slope. We use Ordinary Least Square linear regression to construct a linear model to MASK
175    rainfall and DSS snow accumulation (for the four BoM stations, see Appendix D), and estimate the 95% confidence interval (CI) in the gradient and the intercept as 1.96 multiplied by the standard error of the model.

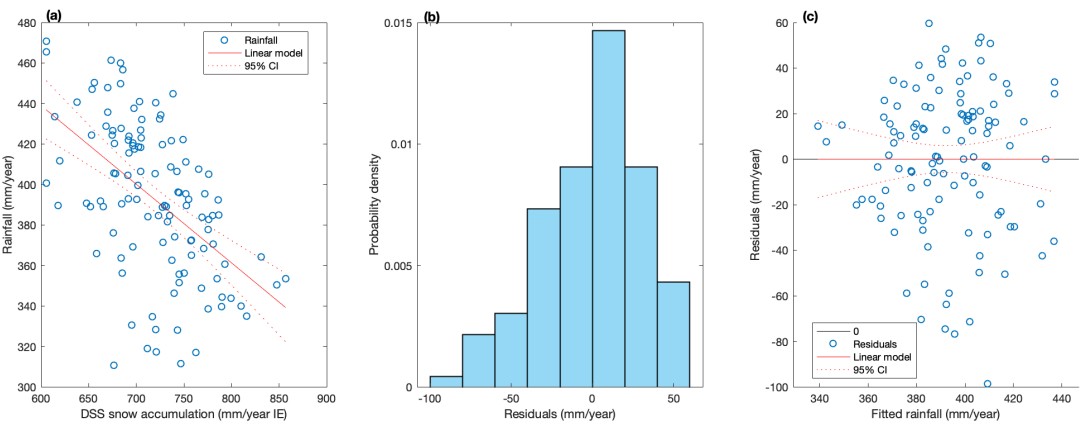

**Figure 2.** (a) The scatter plor for AWAP rainfall in MASK region and DSS snow accumulation (both 6-year window) of period 1900 CE to 2015 CE with their linear model and 95% CI. (b) The histogram for model residuals using probability density function scaling. (c) The scatter plot for model residuals and fitted data with their linear fit and 95% CI.

The model is Rain = Snow * ( - 0.389 ± 0.114 ) + 672 ± 82 mm/year in the growing season for the period 1900 CE to 2015 CE. The gradient interval [-0.503, -0.274] is always negative, which is consistent with the result from the Pearson correlation test, that the coefficient is negative and statistically significant ($p < 0.05$).
180    Figure 2b is a histogram of the raw residuals using probability density function scaling. Residuals broadly follow a normal distribution. Figure 2c is a scatter plot of fitted rainfall data and residuals. The distribution of residuals has no obvious regularity and trend, and is generally symmetric along y = 0. Taken together, there is no obvious evidence to reject the linear model. We use this linear model and the time series of DSS snow accumulation to reconstruct MASK rainfall from 2015 CE back to 22 BCE.





### 3.4 CUSUM analysis

The Cumulative Summation (CUSUM) technique has been used to investigate rainfall data (Kampata et al., 2008; Chowdhury and Beecham, 2010). CUSUM is a method to sum the data anomalies using:

$$y_p = \sum_{k=1}^{p} x_k'$$
(3)

where $x'$ is the anomaly relative to the mean of the whole series (Chowdhury and Beecham, 2010). The aim of CUSUM is to identify the step change in a time series by continuously accumulating anomalies. A step in the original data would be identified by the change of the gradient in the CUSUM.

Numerical integration enhances the signal-to-noise ratio. Using the CUSUM data to both pick the breakpoint (and evaluate its significance) and estimate the timing uncertainty potentially offer advantages compared to using the original data, especially at lower signal-to-noise ratios.

To specifically determine the breakpoint and evaluate its significance, we use the BREAKFIT analysis software (Mudelsee, 2009) on the CUSUM data.

### 3.5 Interpolation and model-data comparison

In light of the different spatial resolutions of the model outputs and AWAP gridded rainfall, we perform the interpolation for model outputs to be the same geospatial resolution as the AWAP gridded rainfall and multiply the MASK matrix with the interpolated model outputs to consistently capture the same region, same geospatial resolution data for both the rainfall reconstruction and simulation.

The rainfall reconstruction of MASK back to 22 BCE is reconstructed by the linear model over MASK where the AWAP gridded rainfall and the DSS snow accumulation are statistically significantly correlated. The rainfall simulation is the precipitation output of the CSIRO Mk3L model run as four ensembles combining four climate forcings (Table 2). We use the CUSUM analysis to compare the output of each ensemble and three members of CONTROL (a pre-industrial control simulation (Phipps et al., 2013)) to the rainfall reconstruction. Also, as each ensemble member represents the randomly different initialised weather states, we compare the mean of each ensemble member to the rainfall reconstruction to estimate the uncertainties in the simulated rainfall.

To statistically compare the rainfall reconstruction with the model outputs and test the differences, we use Welch's t-test to calculate the adapted t-statistics using:

$$t_W = \frac{\bar{x_1} - \bar{x_2}}{\sqrt{\dfrac{s_1^2}{n_1} + \dfrac{s_2^2}{n_2}}}$$
(4)

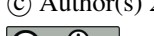


where $\bar{x}_1$ and $\bar{x}_2$, $s_1^2$ and $s_2^2$, $n_1$ and $n_2$ are sample mean values, standard deviations and sample sizes for two samples, respectively. We lalso estimate the adapted degrees of freedom (DF) using the Welch–Satterthwaite equation:

$$\text{DF} = \frac{\left(\dfrac{s_1^2}{n_1} + \dfrac{s_2^2}{n_2}\right)^2}{\dfrac{s_1^4}{n_1^2(n_1-1)} + \dfrac{s_2^4}{n_2^2(n_2-1)}} \tag{5}$$

Taken together, we calculate the p-value using t-statistics and DF.

To evaluate the specific time period we interested in, for example the rainfall before and after 1850 CE, we calculate the Root Mean Square Error (RMSE) between reconstructions with model outputs and also calculate the Pearson correlation coefficients, ESS (Equation 1), Student's t-statistic (Equation 2) and its adjusted p-value.

## 4 Results and discussion

### 4.1 Rainfall reconstruction

Figure 3a shows the reconstructed MASK rainfall time series for the growing season (May to October) from 22 BCE to 2015 CE. This rainfall reconstruction has shown the feasibility of investigating the longer-term context of rainfall variability over SWWA before the instrument-era. We choose 1850 CE to be the year that separates before and after the Industrial Revolution consistent with other studies (e.g. Stocker et al. (2013)).

Next, we assess whether the growing season rainfall in SWWA before and after 1850 CE belong to the same distribution. We divide the rainfall reconstruction into two periods 22 BCE to 1849 CE and 1851 CE to 2015 CE. We plot the empirical distribution function (Figure 4a) and probability density function scaling histogram (Figure 4b) for each sample and its corresponding normal distribution (black dash line) with the same mean and standard deviation.

The distribution functions (Figure 4) for the period 1851 CE to 2015 CE are shifted left relative to the period 22 BCE to 230 1849 CE, indicating the rainfall after 1850 CE has reduced. To quantify the rainfall reduction and test its statistical significance, we perform a two-sample KS test on the two samples and independently conduct Welch's t-test to validate two-sample KS nonparametric test results.

Results from two-sample KS test reject ($p < 0.01$) the null hypothesis that the samples are from same continuous distribution. Independently, Welch's t-test also allows us to reject ($p < 0.01$) the null hypothesis that the means are equal.

The mean of growing season rainfall reconstruction for 1851 CE to 2015 CE is 398 mm/year which is around 98% of the rainfall mean during 22 BCE to 1849 CE which is 405 mm/year. The attenuation of the rainfall before and after 1850 CE is statistically significant but the degree is around 2%. The 1851–2015 CE sample is missing the higher end of the distribution (i.e. there are no years with rainfall greater than 440 mm/year) and this lack of recharging events might have more of an impact than the shift in the mean. Also the 1851 CE–2015 CE sample has a lower low-end of the distribution than 22 BCE–1849 CE, 240 indicating more prevalent drier years. This suggests a possibility of an anthropogenic influence on the hydroclimate of this region.

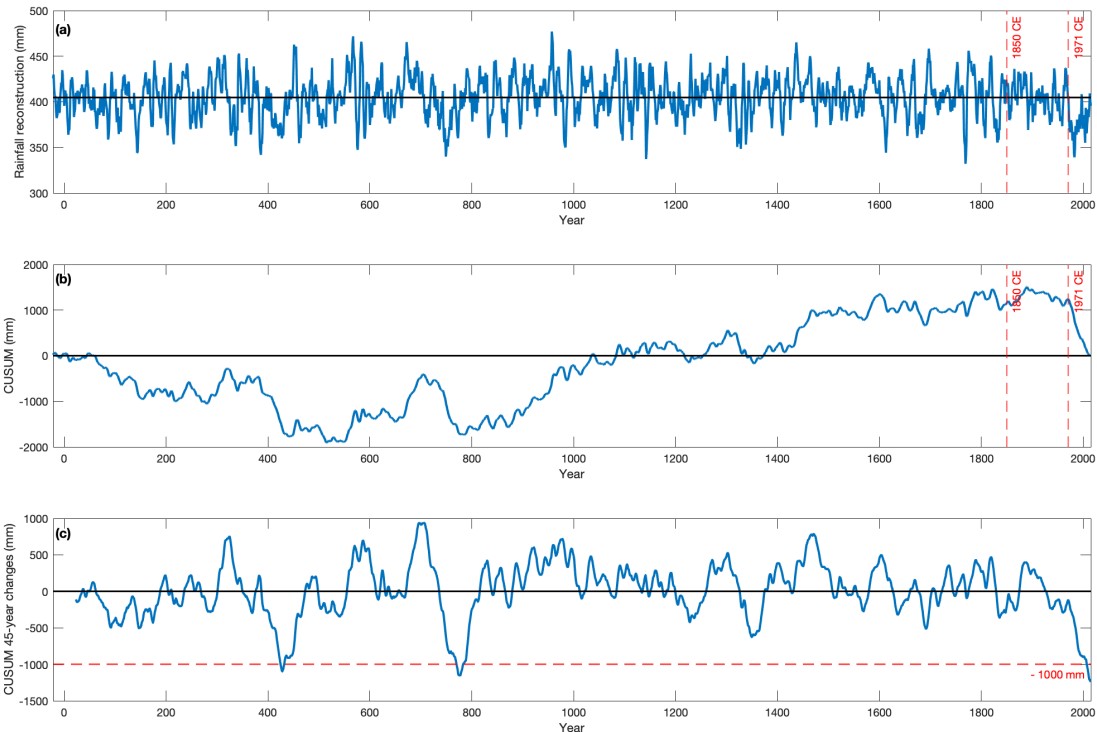

**Figure 3.** (a) The rainfall reconstruction in SWWA significant region (MASK region (Figure 1)) from 22 BCE to 2015 CE. The mean of this 2038-year time series (405 mm/year) is shown as a black horizontal line. (b) The rainfall reconstruction CUSUM (Equation 3). The black horizontal line equals zero. Red vertical dashed lines highlight the years 1850 CE and 1971 CE. (c) The 45-year running change in the rainfall reconstruction CUSUM series. Red horizontal dash line equals -1000 mm (accumulated change).

The time series of the rainfall reconstruction gives us a clue that there might be another attenuation which happened after 1850 CE at around the late 20th century (Figure 3a,b). In order to accurately determine the changes in rainfall after 1850 CE, we calculate the CUSUM time series (Equation 3) for rainfall time series (Figure 3a) to identify the step changes and use BREAK-
FIT analysis on CUSUM to identify any significant changes of the gradient. As CUSUM continuously accumulates anomalies of rainfall reconstruction along timeline, the variability of the rainfall becomes clearly visually observable (Figure 3b). The sign and magnitude of the gradient of CUSUM plot indicate rainfall anomalies. A positive (negative) gradient indicates a positive (negative) anomaly which is a rainfall increase (decrease) event. The larger the magnitude indicates the higher degree of the event. For consistency with the duration of the simulated rainfall which is available from 501 CE to 2000 CE, we take the
rainfall reconstruction for the same period to calculate CUSUM time series and conduct BREAKFIT analysis.





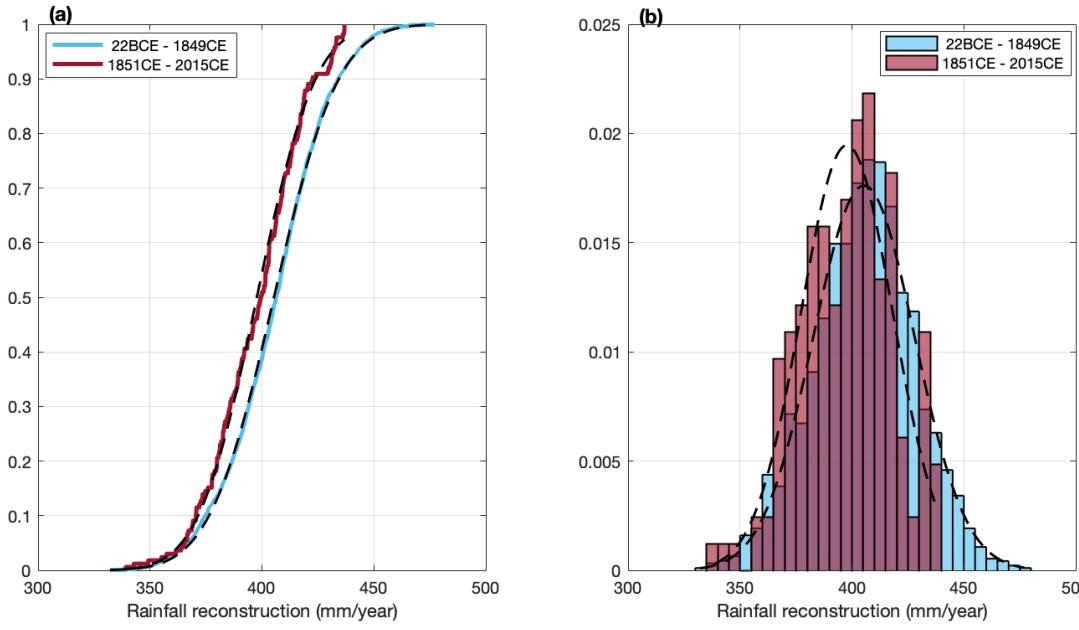

**Figure 4.** (a) Empirical distribution function plot for rainfall reconstruction samples 22 BCE to 1849 CE (blue curve) and 1851 CE to 2015 CE (red curve). (b) Probability density function scaling histogram for two samples. Black dash curves are each sample's corresponding normal distribution.

Figure 5 shows the rainfall reconstruction CUSUM time series from 1850 CE to 2000 CE subsampled from the 501 CE to 2000 CE series. Results from BREAKFIT show the break-point is at 1971 CE $\pm$ 7 years (95% CI). The gradients for the intervals (1850, 1971) CE and (1971, 2000) CE are -1.4 mm/year [-3.2,-0.4] and -37.16 mm/year [-45.9,-28.8] respectively. Thus, the break in gradient at 1971 is statistically significant.

We test the findings of break-point analysis on the CUSUM time series by analysis of the original data. Breaking the rainfall into two samples, we found a shift of distribution to lower rainfall after 1971 CE in both empirical distribution function (Figure 6a) and probability density function scaling histogram (Figure 6b). The shift is around 30 mm/year showing a nearly 8% attenuation after 1971 CE, more than 4 times the reduction from 1850 CE. Two further statistical tests were preformed to verify this finding.

Results from the two-sample KS test allow us to reject the null hypothesis (p<0.01) that the samples are from the same continuous distribution. Independently, it is statistically significant (p < 0.01) that the result from Welch's t-test can also allow us to reject the null hypothesis that two samples have equal means. The mean rainfall during the period 1972 CE to 2000 CE is 373 mm/year, which is 92% of the mean rainfall during the period 1850 CE to 1970 CE (406 mm/year). This change is more than four times bigger than the change around 1850 CE. Not only is there a large reduction in the mean rainfall, but also the

1972 CE to 2000 CE sample's distribution has largely shifted to lower values compared with 1850 CE to 1970 CE. There are





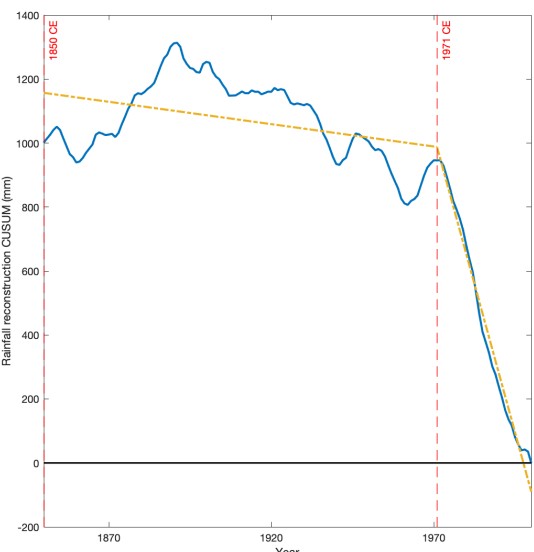

**Figure 5.** CUSUM time series on rainfall reconstruction in period 1850–2000 CE from the 501–2000 CE series. Yellow dotted lines are each interval's gradient calculated by using BREAKFIT analysis, with 1971 CE identified as the break-point. The gradient before 1971 CE is -1.40 mm/year. The gradient after 1971 CE is -37.16 mm/year.

no years with rainfall over 410 mm/year, and nearly half of the years have rainfall less than 375 mm/year in the 1972 CE to 2000 CE sample (Figure 6b). This likely has a larger impact on agriculture than the rainfall shift before and after 1850 CE.

Results from the two individual statistical tests are very consistent with the BREAKFIT analysis on CUSUM time series that indicate 1971 CE marks the change in gradients from 1850 CE to 2000 CE. Comparison of Figure 6 with Figure 4 reveals
that the shift in the rainfall distribution at around 1971 CE was much more pronounced than the shift at around 1850 CE. The reduction in the mean rainfall at around 1971 CE resulted in a prolonged drought in SWWA. This drought might be continuing as we do not consider the period after 2015 CE in this study. To highlight dry epochs of an equivalent duration to the observed drought to date, we calculate the 45-year running change in the CUSUM series (Figure 3c). Over the 45 years from 1971 CE to 2015 CE, this prolonged drought is expressed by an integrated rainfall reduction of more than 1000 mm (Figure 3c). We
note there are two comparable prior epochs during the past two millennia, lasting from around 385–429 CE and 732–776 CE (Figure 3c).

## 4.2  Data-model comparison

Consistent with the analysis applied to the rainfall reconstruction, we analyze the simulated rainfall from CONTROL and the four forced ensembles. We divide the simulated time series for each simulation into two samples: 501 CE to 1849 CE and
1851 CE to 2000 CE. We further divide the industrial period into samples before and after the observed shift in 1971 CE. We





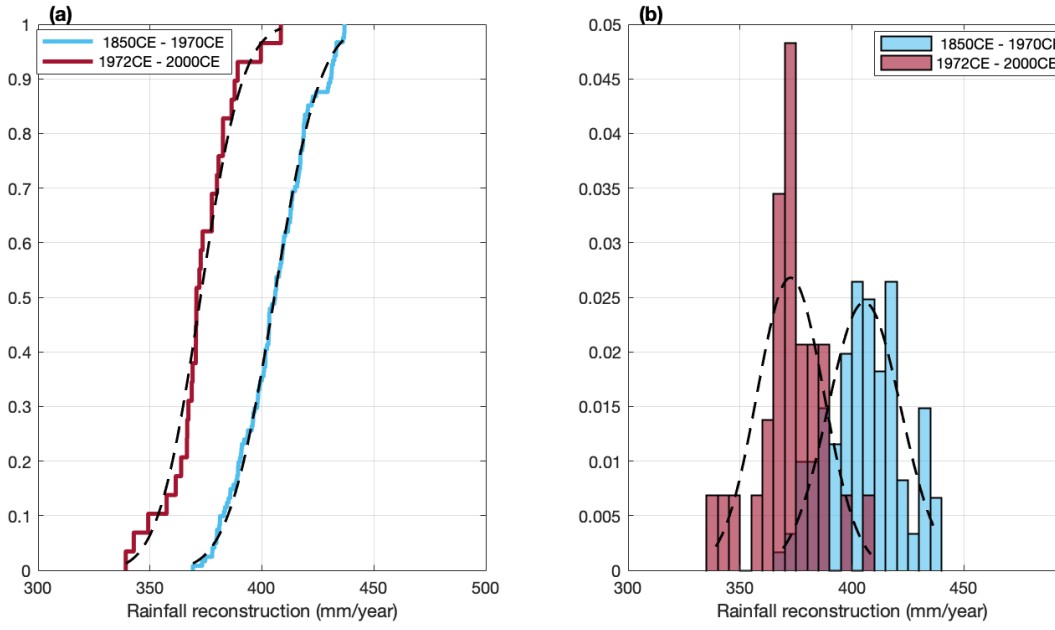

**Figure 6.** (a) Empirical distribution function plot for rainfall reconstruction samples 1850 CE to 1970 CE (blue curve) and 1972 CE to 2000 CE (red curve). (b) Probability density function scaling histogram for two samples. Black dash curves are each sample's corresponding normal distribution.

then independently perform the two-sample KS test and the Welch's t-test. We also use two different metrics to assess the degree of agreement between the CUSUM time series for the various model simulations and the reconstruction: the differences in the slope, and the Root Mean Square Error (RMSE).

Summary statistics for the model simulations are shown in Table E1. The model can be seen to have a dry bias, but the
simulated internal variability is of a similar magnitude ($\sim 80\%$) to the reconstruction. The simulated rainfall for each individual ensemble member is shown in Figure E1, with the corresponding CUSUM time series shown in Figure E2. Within each forced ensemble, there are considerable differences between the CUSUM time series for individual ensemble members. This highlights the role of unforced internal variability in driving SWWA rainfall, consistent with the findings of Cai and Shi (2005). The 45-year changes in the CUSUM for the pre-industrial control simulation are shown in Figure E3. A number of prolonged
droughts that approach the magnitude of the current integrated rainfall deficit are apparent, suggesting that the prior drought epochs featuring in the reconstruction may have arisen through natural climate variability or natural forcings such as volcanoes.

### 4.2.1 Industrial era

We now examine the model ensembles and assess whether the simulated rainfall during the industrial era has a different distribution from the simulated rainfall during the pre-industrial era. We cannot reject the null hypothesis, with either the



**Table 4.** The results for two independent statistical tests for different time periods from the rainfall time series. KS test is the two-sample KS test and W's t-test is the Welch's t-test. The tests results "same" means we cannot reject the null hypothesis (the two samples of data are from the same distribution), while "different" means we can reject the null hypothesis (p < 0.05).

|  | 501–1849 CE VS 1851–2000 CE | | 1850–1970 CE VS 1972–2000 CE | |
| --- | --- | --- | --- | --- |
| Sample | KS test | W's t-test | KS test | W's t-test |
| CONTROL | same | same | same | same |
| O | same | same | same | same |
| OG | different | different | same | same |
| OGS | different | different | same | same |
| OGSV | different | different | same | same |
| Rainfall reconstructions | different | different | different | different |

two-sample KS test or Welch's t-test performed on both CONTROL and O series, that the rainfall in 501–1849 CE and 1851–2000 CE are from the same distribution (KS test) or equal mean (Welch's t-test) (Table 4). Conversely, the results from both two-sample KS test and Welch's t-test allow us to reject (p < 0.05) the null hypothesis for OG, OGS or OGSV and rainfall reconstruction (Table 4). These results give us confidence to say that orbital forcing is not the main driver for the rainfall changes after 1850 CE. Once the greenhouse gases are added to the model climate forcings, the rainfall simulations before and

after 1850 CE belong to different distributions with different means (Table 4). There is no change when the solar irradiance and volcanic aerosols are added. Based on these statistical tests, we suggest that there is evidence for a role of anthropogenic greenhouse gases. However, there is no evidence of any additional impacts due to either solar irradiance or volcanic aerosols.

Figure 7 shows the CUSUM time series for each of the rainfall reconstruction and CSIRO Mk3L model simulations from 501 CE to 2000 CE. For each ensemble including CONTROL scenario, the CUSUM time series is the mean of three members.

As for the statistical tests (Table 4), the CUSUM series in Figure 7 show that the rainfall change around 1850 CE is not simulated by CONTROL or O ensembles, but is simulated by OG, OGS and OGSV ensemble series.

Examining each of the ensembles in turn, we see that CONTROL does not capture the key features of the the rainfall reconstruction (Figure 7). This suggests either a role of internal variability, or that climate forcings might be driving the rainfall changes in SWWA. The time series of O varies slightly around zero with no obvious gradient changes (Figure 7), which

suggests that orbital forcing alone cannot explain the changes in SWWA rainfall in recent decades. In contrast, ensembles OG, OGS and OGSV are able to reproduce some of the key features of the rainfall reconstruction, particularly the decline after 1850 CE. These ensembles have approximately equal magnitudes especially after around 1900 CE and their gradient changes are broadly synchronous with large negative magnitudes.





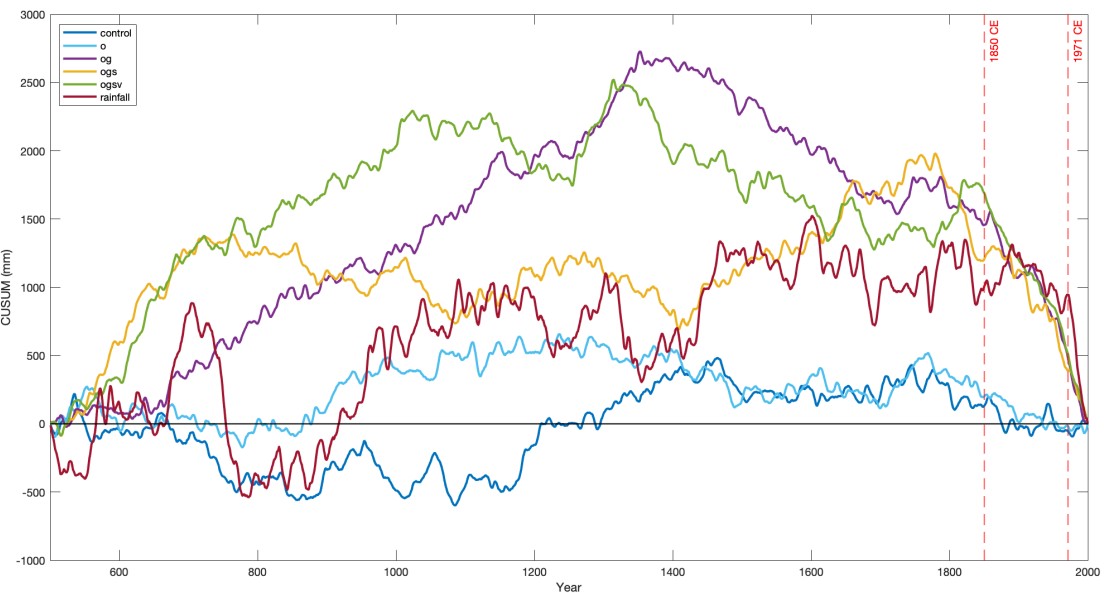

**Figure 7.** CUSUM time series for the rainfall reconstruction and CSIRO Mk3L model rainfall simulations in the MASK region from 501 CE to 2000 CE. The model simulations include CONTROL (pre-industrial control simulation (Phipps et al., 2013)), O, OG, OGS and OGSV. The black horizontal line equals zero. Red vertical dash lines highlight the year 1850 CE and 1971 CE.

### 4.2.2    Late 20th century

Figure 8 shows the CUSUM time series for the rainfall reconstruction and simulations in the period 1972–2000 CE. Ensemble O shows that the orbital forcing cannot explain the rainfall reduction after 1971 CE. However the reduction is reproduced by ensembles OG, OGS and OGSV. All three ensembles have the same sign of gradients and similar magnitudes. The magnitude of the gradient for the rainfall reconstruction is larger than any of the rainfall simulations.

We also perform a two-sample KS test and Welch's t-test on the rainfall simulations in the periods 1850–1970 CE and 1972–
2000 CE. None of the model results from the statistical tests allow us to reject the null hypothesis with statistical significance. Unlike the rainfall reconstruction, none of the model simulations show that the rainfall in period 1972–2000 CE is different from the period 1850–1970 CE (Table 4).

As we can see in Figure 8, CONTROL and O have negligible slope. However, OG, OGS and OGSV do exhibit a negative slope and are therefore much closer to the rainfall reconstruction. We note that the amplitude of the internal variability in the
underlying raw time series will influence the magnitude of the slopes in the CUSUM analysis. CONTROL and O are essentially indistinguishable as the time frame of orbital forcing is long compared to the 30 year period studied here. To determine which





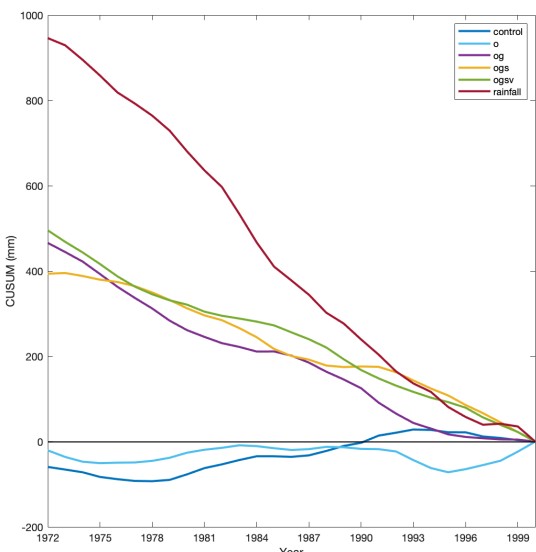

**Figure 8.** The CUSUM time series on the rainfall reconstruction and simulations in the period 1972–2000 CE from the 501–2000 CE series. The black horizontal line equals zero.

**Table 5.** The RMSE and difference in slope calculated between the rainfall reconstruction and simulations in the period 1972–2000 CE. Sample represents to each rainfall simulations. The difference in slope is the difference of the slope of the Ordinary Least Square linear fitted CUSUM time series data for each sample to the slope of the fitted CUSUM time series data for rainfall reconstruction in the period 1972–2000 CE.

| Simulation | Difference in slope (mm/year) | RMSE (mm) |
| --- | --- | --- |
| CONTROL | 41.50 | 580 |
| O | 37.00 | 558 |
| OG | 20.09 | 296 |
| OGS | 23.00 | 287 |
| OGSV | 20.66 | 263 |

model simulation is in best agreement with the rainfall reconstruction, we calculate the difference in slope and the Root Mean Square Error (RMSE) (Table 5) between the CUSUM series for the rainfall reconstruction and each model ensemble.

Comparing CONTROL and the four ensembles, CONTROL and O have the largest differences in slope and RMSE, relative to the rainfall reconstruction, than any of the others. The forced ensembles OG, OGS and OGSV have much smaller differences in slope and much smaller RMSE. Therefore, both of difference in slope and RMSE analysis suggest that the simulations





including greenhouse gases are the closest to the rainfall reconstruction. Including solar irradiance and volcanic aerosols has negligible additional impact.

Although the model ensembles that include greenhouse gas forcing do simulate a drying trend after 1971 CE, the simulated
trend is weaker than in the reconstruction. This may reflect the influence of stratospheric ozone depletion, which is not modelled in the climate model simulations that we analyse in this study. Alternatively, it may reflect the role of natural climate variability. Because of its stochastic nature, the model simulations would not be expected to replicate the phase or magnitude of the specific internal variability encountered in the real world.

The largest improvement in agreement comes from adding greenhouse gases into the model. Therefore, our analysis suggests
that greenhouse gases are the dominant climate forcing out of the four natural and anthropogenic forcing considered in this study driving the rainfall changes in SWWA during the 20th century.

## 5    Conclusions

We have used the DSS snow accumulation record to extend the previous published reconstruction of SWWA growing season (May to October) rainfall. Based on the statistically significant (p < 0.05) negative correlation between MASK rainfall and
DSS snow accumulation in the growing season, we built a linear model and reconstructed the rainfall for 2038 years (22 BCE to 2015 CE). Consistent with other studies (e.g. Stocker et al. (2013)), we use 1850 CE to divide the reconstruction into two periods: before and after the industrial revolution. We independently performed the two-sample KS test and Welch's t-test to test whether the two samples are from the equal means and distributions. The results from tests rejected (p < 0.01) the null hypotheses, suggesting a detectable change in SWWA rainfall much earlier than the observed climate shift in SWWA in
the 1970s. The CUSUM time series and the BREAKFIT analysis suggest a reduction rainfall after 1971 CE that was much larger in magnitude than the reduction after 1850 CE. The rainfall means and distributions in the periods 1850–1970 CE and 1972–2000 CE are independently tested and shown to be statistically significantly (p < 0.01) different.

With statistically significant (p < 0.01) rainfall reductions after 1850 CE and after 1971 CE, we compared the rainfall reconstructions with ensembles of CSIRO Mk3L climate model simulations driven by different natural and anthropogenic climate
forcings. Comparing the periods 501–1849 CE and 1851–2000 CE, adding greenhouse gases significantly (p < 0.05) changed the rainfall means and distributions for the two periods. There is no detectable influence from solar irradiance and volcanic aerosols. However, the rainfall means and distributions for the periods 1850–1970 CE and 1972–2000 CE are statistically indistinguishable.

Examining the CUSUM for the period 1972–2000 CE, we do not find any drying trend in CONTROL simulation or in
simulations driven only with orbital forcing. However, we do find a drying trend in simulations that include anthropogenic greenhouse gases. We again find that solar irradiance and volcanic aerosols have negligible additional impact. The model simulations therefore suggest that anthropogenic forcing has been the dominant driver of the rainfall trend in SWWA since 1971 CE.



Both the reconstruction and the climate model simulations suggest that the drying trend began earlier than the 1970s. How-
ever, the model simulations do not capture the acceleration in the reconstructed drying trend at around 1971 CE. This suggests
that this acceleration cannot be attributed to external forcings, or at least not any of the forcings considered in this study. Either
natural variability or stratospheric ozone depletion are potential alternative explanations (e.g. Cai and Shi, 2005).

The reconstruction reveals that the rainfall reduction in SWWA in recent decades is not unprecedented. We note that there
are two previous prolonged drought events of similar magnitude in SWWA during the past two millennia. Droughts of similar
duration and magnitude occur in an unforced pre-industrial control simulation, suggesting that these events may have arisen
through natural climate variability.





*Data availability.* The SWWA rainfall reconstruction data is available via https://doi.org/10.25959/5f4c50b7b661f. The DSS snow accu-
mulation record is available via https://www.clim-past.net/11/697/2015/. The CSIRO Mk3L climate model simulations are available via
https://doi.org/10.5281/zenodo.3908926/. The BoM instrumental rainfall records are available via http://www.bom.gov.au/climate/data/. The
AWAP AGCD data is available via http://www.bom.gov.au/metadata/catalogue/19115/ANZCW0503900567?template=full.

## Appendix A: Tests for normality

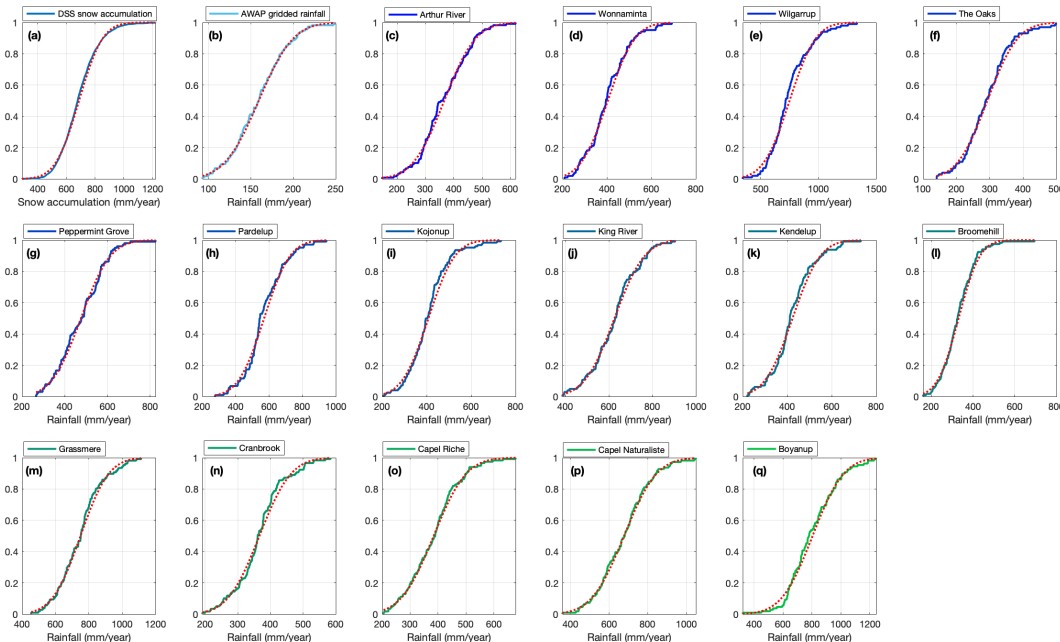

**Figure A1.** The CDF of observational data for testing normality. Curves are empirical CDF of DSS snow accumulation, AWAP gridded
rainfall and 15 BoM stations rainfall data, red dotted lines are CDF for normal distribution corresponding to each sample's empirical CDF
with equal mean and standard deviation, respectively. For each panel, (a) is the DSS snow accumulation, (b) is the AWAP gridded rainfall,
(c) to (q) are each of the 15 BoM stations rainfall, respectively.

Figure A1 shows CDFs for DSS snow accumulation, AWAP gridded rainfall and 15 BoM stations rainfall. All of these 17
CDFs are in agreement with their corresponding normal distributions with negligible biases. Not only are the shapes between
the empirical and normal CDFs similar, but also the distances between them are negligible. This visual validation has shown
consistency with the results from one-sample KS tests that we cannot reject the null hypothesises that the data comes from nor-
mal distributions. Therefore, all of the rainfall data passes the normality tests. Therefore, we can apply the Pearson correlation
coefficients and also perform parametric tests on any of the rainfall data.





## Appendix B: Defining the window size for smoothing

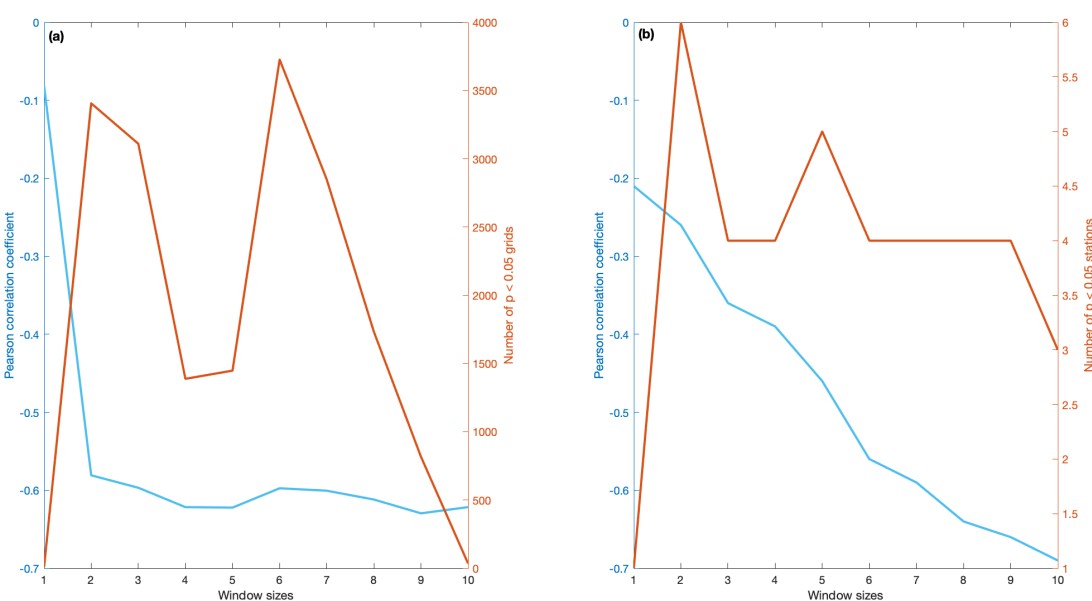

**Figure B1.** (a) Pearson correlation coefficient and the number of the statistically significant ($p < 0.05$) grids for AWAP rainfall data under different window sizes of moving average smoothing. (b) Pearson correlation coefficient and the number of the statistically significant ($p < 0.05$) stations for 15 BoM stations rainfall data under different window sizes of moving average smoothing. For both panels, blue lines and the left-hand side y-axis are the average of the Pearson correlation coefficient of the statistically significant ($p < 0.05$) grids/stations, red lines and the right-hand side y-axis are for the number of the statistically significant ($p < 0.05$) grids/stations.

We test the effect of smoothing, using window sizes of 1–10 years, on the correlation between SWWA rainfall and DSS
accumulation data. We calculate the Pearson correlation coefficient, estimate the ESS (Equation 1) and calculate Student's t-statistic (Equation 2) for each window of the AWAP gridded rainfall and BoM stations rainfall. Figure B1(a) is the results of the Pearson correlation coefficient and the number of the grids that have statistically significant ($p < 0.05$) correlations for the AWAP gridded rainfall. A six-year window maximizes the number of the grids (Figure B1a) with statistically significant ($p < 0.05$) correlations, and negligibly changes the magnitude of the correlation coefficient compared with the adjacent window
sizes (Figure B1a). Therefore, a six-year window is the optimal window size for moving average smoothing on the AWAP gridded rainfall.

The results of the 15 BoM stations show differences compared with the AWAP gridded rainfall. A two-year window maximizes the number of the stations (Figure B1b) with statistically significant ($p < 0.05$) correlations. But the correlation is relatively low compared with the larger window sizes (Figure B1b). The correlation increases as the window size increases




(Figure B1b). For consistency with the AWAP gridded rainfall, we also use six-year window to smooth the BoM stations
rainfall.

## Appendix C:  Significance testing for 110 Local Government Areas

Table C1: The table for all 110 Local Government Areas (LGAs) with their Pearson correlation coefficients ($r_6$) and ESS
($N_6^{eff}$) (under 6-year moving average smoothing). For some LGAs, "−" means the area of this LGA is less than one grid area
of the geospatial resolution of the AWAP data. The bold numbers of correlation are statistically significant (p < 0.05).

| Local Government Area | $r_6$ | $N_6^{eff}$ | Local Government | $r_6$ | $N_6^{eff}$ |
|---|---|---|---|---|---|
| Albany | -0.35 | 0.17 | Dalwallinu | **-0.46** | 0.17 |
| Armadale | **-0.51** | 0.13 | Dandaragan | -0.44 | 0.14 |
| Augusta Margaret River | **-0.48** | 0.15 | Dardanup | **-0.54** | 0.14 |
| Bassendean | **-0.48** | 0.15 | Denmark | -0.44 | 0.16 |
| Bayswater | -0.45 | 0.15 | Donnybrook Balingup | **-0.53** | 0.15 |
| Belmont | -0.47 | 0.14 | Dowerin | -0.43 | 0.18 |
| Beverley | **-0.50** | 0.15 | Dumbleyung | -0.41 | 0.15 |
| Boddington | **-0.51** | 0.15 | Dundas | -0.31 | 0.21 |
| Boyanup Brook | **-0.53** | 0.17 | East Fremantle | – | – |
| Bridgetown Greenbushes | **-0.51** | 0.16 | Fremantle | -0.38 | 0.14 |
| Brookton | **-0.49** | 0.15 | Gingin | -0.46 | 0.14 |
| Broomehill Tambellup | -0.45 | 0.15 | Gnowangerup | -0.42 | 0.15 |
| Bruce Rock | **-0.46** | 0.17 | Goomalling | **-0.45** | 0.17 |
| Bunbury | **-0.55** | 0.14 | Gosnells | -0.48 | 0.14 |
| Busselton | **-0.49** | 0.15 | Greater Geraldton | -0.36 | 0.16 |
| Cambridge | – | – | Harvey | **-0.51** | 0.15 |
| Canning | -0.46 | 0.14 | Inwin | **-0.48** | 0.15 |
| Capel | **-0.55** | 0.14 | Joondalup | -0.47 | 0.14 |
| Carnamah | **-0.48** | 0.15 | Kalamunda | **-0.50** | 0.14 |
| Chapman Valley | -0.39 | 0.16 | Katanning | **-0.51** | 0.15 |
| Chittering | **-0.50** | 0.14 | Kellerberrin | -0.42 | 0.19 |
| Claremont | – | – | Kent | -0.45 | 0.15 |
| Cockburn | -0.42 | 0.14 | Kojonup | **-0.47** | 0.15 |
| Collie | **-0.53** | 0.15 | Kondinin | **-0.44** | 0.19 |
| Coorow | -0.46 | 0.15 | Koorda | -0.38 | 0.17 |





| | | | | | |
|---|---|---|---|---|---|
| Corrigin | -0.45 | 0.16 | Kulin | -0.43 | 0.18 |
| Cottsloe | – | – | Kwinana | -0.46 | 0.14 |
| Cranbrook | **-0.51** | 0.16 | Lake Grace | -0.45 | 0.16 |
| Cuballing | **-0.48** | 0.15 | Mandurah | -0.42 | 0.14 |
| Cunderdin | **-0.47** | 0.18 | Manjimup | **-0.54** | 0.16 |
| Melville | -0.42 | 0.14 | South Perth | – | – |
| Merredin | **-0.42** | 0.20 | Stirling | -0.45 | 0.15 |
| Mingenew | **-0.49** | 0.16 | Subiaco | – | – |
| Moora | **-0.48** | 0.16 | Swan | **-0.49** | 0.15 |
| Morawa | -0.39 | 0.16 | Tammin | **-0.46** | 0.18 |
| Mosman Park | – | – | Three Springs | **-0.51** | 0.15 |
| Mount Marshall | -0.40 | 0.16 | Toodyay | -0.46 | 0.15 |
| Mukinbudin | -0.42 | 0.17 | Trayning | -0.36 | 0.19 |
| Mundaring | **-0.48** | 0.15 | Victoria Park | -0.44 | 0.14 |
| Murray | **-0.49** | 0.14 | Victoria Plains | **-0.50** | 0.15 |
| Nannup | **-0.52** | 0.15 | Vincent | – | – |
| Narembeen | **-0.43** | 0.19 | Waggin | -0.43 | 0.16 |
| Narrogin | -0.45 | 0.16 | Wandering | **-0.53** | 0.14 |
| Nedlands | -0.41 | 0.14 | Wanneroo | -0.48 | 0.14 |
| Northam | -0.42 | 0.16 | Waroona | **-0.49** | 0.15 |
| Northampton | -0.36 | 0.15 | West Arthur | **-0.54** | 0.15 |
| Nungarin | **-0.41** | 0.20 | Westonia | -0.34 | 0.20 |
| Peppermint Grove | – | – | Wickepin | -0.45 | 0.15 |
| Perenjori | -0.43 | 0.15 | Williams | **-0.53** | 0.15 |
| Perth | -0.42 | 0.15 | Wongan Ballidu | **-0.45** | 0.17 |
| Pingelly | **-0.49** | 0.15 | Woodaniling | **-0.49** | 0.15 |
| Plantegenet | **-0.47** | 0.16 | Wyalkatchem | -0.41 | 0.18 |
| Quairading | **-0.46** | 0.17 | Yalgoo | -0.18 | 0.17 |
| Rockingham | -0.44 | 0.14 | Yilgran | -0.20 | 0.19 |
| Serpentine Jarrahdale | **-0.52** | 0.14 | York | **-0.47** | 0.15 |

We independently calculate the Pearson correlation coefficients and test their statistical significance for each Local Government Areas (LGAs), and then combine the areas which are statistically significant ($p < 0.05$) to make the "MASK" region. Table C1 shows the results for each LGA. The rainfall data for each LGA is smoothed by using 6-year moving average smoothing






(see Appendix B about how to define the window size). The ESS ($N_6^{eff}$) and Student's t-statistic (for testing the statistical significance) are calculated by using Equation 1 and Equation 2.

**Appendix D:  Significance testing and linear model construction for BoM stations**

**Table D1.** The ID and location for each of the 15 BoM rainfall stations and their rainfall record periods with the calculated results of Pearson correlation coefficients and the estimated ESS. $r$ and $N^{eff}$ are Pearson correlation coefficients and ESS for raw data. $r_6$ and $N_6^{eff}$ are Pearson correlation coefficients and ESS for moving average smoothed data with 6-year window size.

| Station | ID | Longitude | Latitude | r | $N^{eff}$ | $r_6$ | $N_6^{eff}$ | Period (CE) |
|---|---|---|---|---|---|---|---|---|
| Arthur River | 10505 | 117.03 °E | 33.34 °S | -0.11 | 121 | **-0.55** | 19 | 1891–2015 |
| Boyanup | 9503 | 115.73 °E | 33.48 °S | -0.18 | 114 | **-0.62** | 16 | 1898–2013 |
| Broomehill | 10525 | 117.64 °E | 33.85 °S | 0.01 | 122 | -0.35 | 21 | 1891–2015 |
| Capel Naturaliste | 9519 | 115.02 °E | 33.54 °S | -0.16 | 110 | -0.40 | 16 | 1904–2015 |
| Cranbrook | 10537 | 117.57 °E | 34.30 °S | -0.08 | 123 | **-0.54** | 19 | 1891–2015 |
| Capel Riche | 9520 | 118.75 °E | 34.61 °S | -0.12 | 117 | -0.01 | 19 | 1897–2015 |
| Grassmere | 9551 | 117.76 °E | 35.02 °S | -0.07 | 109 | -0.39 | 16 | 1903–2015 |
| Kendelup | 9561 | 117.63 °E | 34.49 °S | -0.15 | 113 | -0.42 | 14 | 1901–2015 |
| King River | 9564 | 117.92 °E | 34.94 °S | **-0.21** | 102 | -0.35 | 15 | 1904–2007 |
| Kojonup | 10582 | 117.15 °E | 33.84 °S | 0.01 | 127 | -0.32 | 19 | 1885–2015 |
| Pardelup | 9591 | 117.38 °E | 34.64 °S | -0.04 | 114 | -0.46 | 17 | 1900–2015 |
| Peppermint Grove | 9594 | 119.36 °E | 34.44 °S | -0.18 | 107 | -0.13 | 31 | 1904–2015 |
| The Oaks | 10636 | 117.67 °E | 33.16 °S | 0.02 | 104 | -0.31 | 17 | 1907–2012 |
| Wilgarrup | 9619 | 116.02 °E | 34.15 °S | -0.11 | 112 | -0.40 | 16 | 1901–2014 |
| Wonnaminta | 10658 | 116.99 °E | 32.83 °S | -0.15 | 109 | **-0.55** | 16 | 1905–2015 |

**Table D2.** The gradients and intervals of the linear models of each of the four BoM stations rainfall, with RMSE and periods.

| Sample | Gradient | Interval | RMSE | Period (CE) |
|---|---|---|---|---|
| Arthur River | [-0.63, -0.36] | [627, 821] | 40.0 mm | 1891–2015 |
| Boyanup | [-1.60, -1.00] | [1517, 1952] | 83.5 mm | 1898–2013 |
| Cranbrook | [-0.55, -0.31] | [588, 759] | 35.1 mm | 1891–2015 |
| Wonnaminta | [-0.64, -0.35] | [657, 864] | 39.7 mm | 1905–2015 |





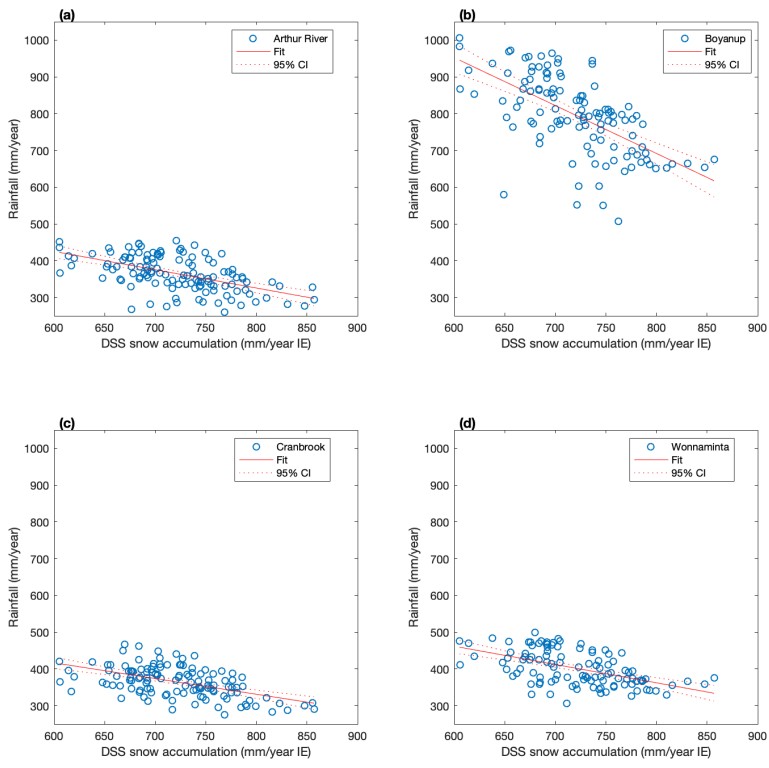

**Figure D1.** The scatter plor for four BoM stations rainfall that have statistically significant correlation (6-year smoothing, p < 0.05) and their linear model with 95% CI. For each panel, (a) is the Arthur River station, (b) is the Boyanup station, (c) is the Cranbrook station and (d) is the Wonnaminta station.

Table D1 shows the results of significance testing for each of the 15 BoM rainfall stations. King River shows the statistically

significant (p < 0.05) correlation on the raw data. We have discussed in Appendix B that 5-year window maximizes the number of the stations with statistically significant (p < 0.05) correlations. For consistence with AWAP gridded rainfall, we here also calculate and test the 6-year smoothing rainfall for 15 BoM stations data. With 6-year moving average smoothing, the number of the stations that show statistical significance (p < 0.05) rises to four, and also the correlations become stronger.

Figure D1 shows the scatter plot for each BoM stations which have statistically significant (p < 0.05, 6-year smoothing)

correlation. The distributions of data all show negative trends. We also build linear models for each station. Table D2 shows the gradients, intervals and RMSE for each stations. For each of these four stations, the gradient is always negative (Table D2). Negative gradients show consistency with the results of the Pearson correlation coefficients that the $r_6$ for each of these four stations are statistically significant (p < 0.05) negative (Table D1), and also the negative trend distributions for these four stations (Figure D1).





**Appendix E: CUSUM analysis on each member of the model ensembles**

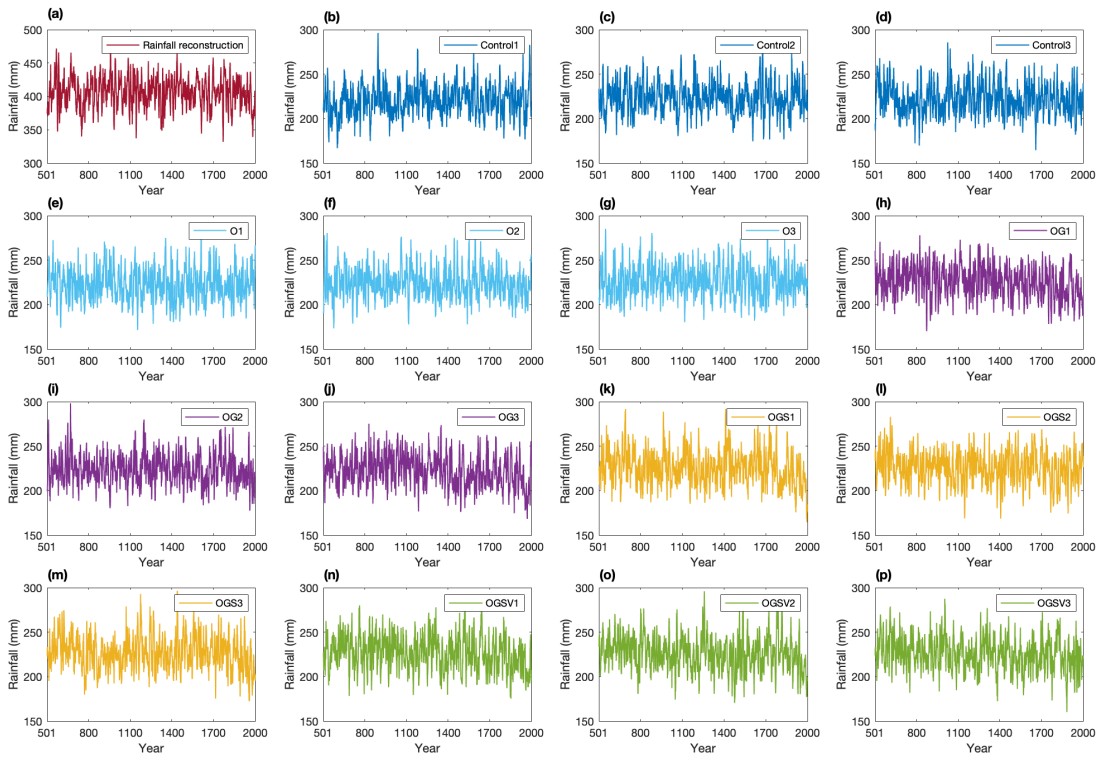

**Figure E1.** Growing season rainfall for SWWA rainfall reconstruction and each member of the CSIRO M3L model rainfall simulations in the MASK region from 501 CE to 2000 CE. For each panel: (a) is the rainfall reconstruction, (b), (c) and (d) are the each member of CONTROL simulations, (e), (f) and (g) are the each member of the O (orbital forcing (Table 2)) simulations, (h), (i) and (j) are the each member of the OG (orbital and greenhouse gases (Table 2)) simulations, (k), (l) and (m) are the each member of the OGS (orbital, greenhouse gases and solar irradiance (Table 2)) simulations, (n), (o) and (p) are the each member of the OGSV (orbital, greenhouse gases, solar irradiance and volcanic aerosols (Table 2)) simulations.





**Table E1.** The mean and standard deviation for each of the growing season rainfall time series from 501 CE to 2000 CE.

| Time series | Mean | Standard deviation |
|---|---|---|
| Rainfall reconstruction | 406.00 | 22.61 |
| CONTROL1 | 220.59 | 17.39 |
| CONTROL2 | 223.75 | 17.16 |
| CONTROL3 | 221.96 | 17.40 |
| O1 | 223.21 | 17.95 |
| O2 | 224.12 | 16.79 |
| O3 | 226.60 | 17.47 |
| OG1 | 226.01 | 17.86 |
| OG2 | 224.21 | 16.95 |
| OG3 | 222.56 | 17.78 |
| OGS1 | 226.26 | 18.96 |
| OGS2 | 226.16 | 17.74 |
| OGS3 | 226.99 | 18.69 |
| OGSV1 | 226.98 | 19.08 |
| OGSV2 | 226.26 | 19.38 |
| OGSV3 | 225.63 | 17.97 |



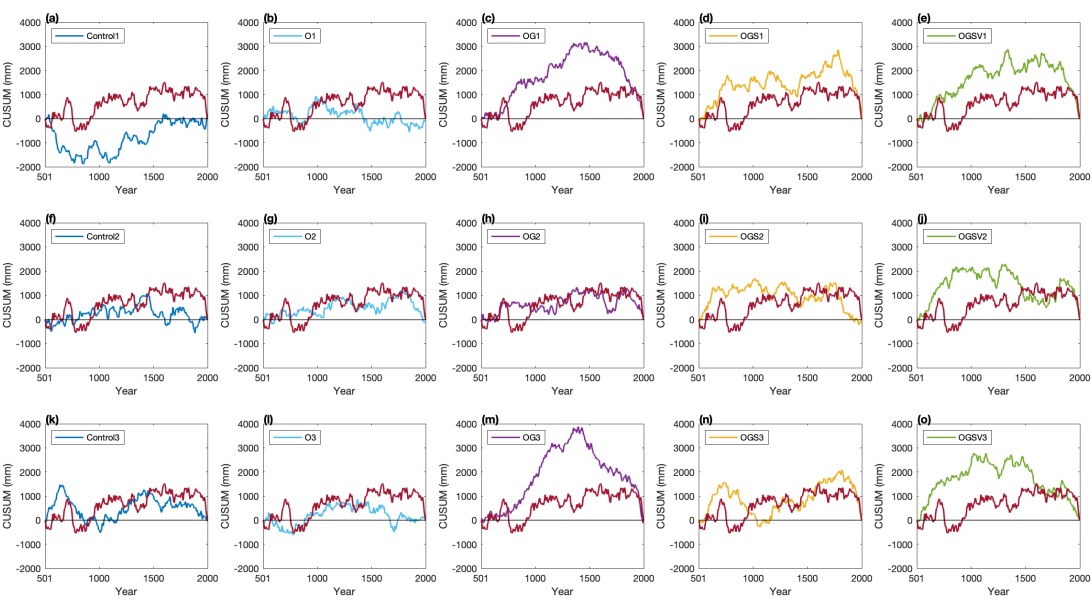

**Figure E2.** CUSUM time series for each member of the CSIRO M3L model rainfall simulations with the rainfall reconstruction in the MASK region from 501 CE to 2000 CE. For each panel: (a), (f) and (k) are the each member of CONTROL simulations with the rainfall reconstruction, (b), (g) and (l) are the each member of the O simulations with rainfall reconstruction, (c), (h) and (m) are the each member of the OG simulations with rainfall reconstruction, (d), (i) and (n) are the each member of the OGS simulations with rainfall reconstruction, (e), (j) and (o) are the each member of the OGSV simulations with rainfall reconstruction. For each panel, the red curve is the CUSUM time series for rainfall reconstruction, and the black horizontal line equals zero.




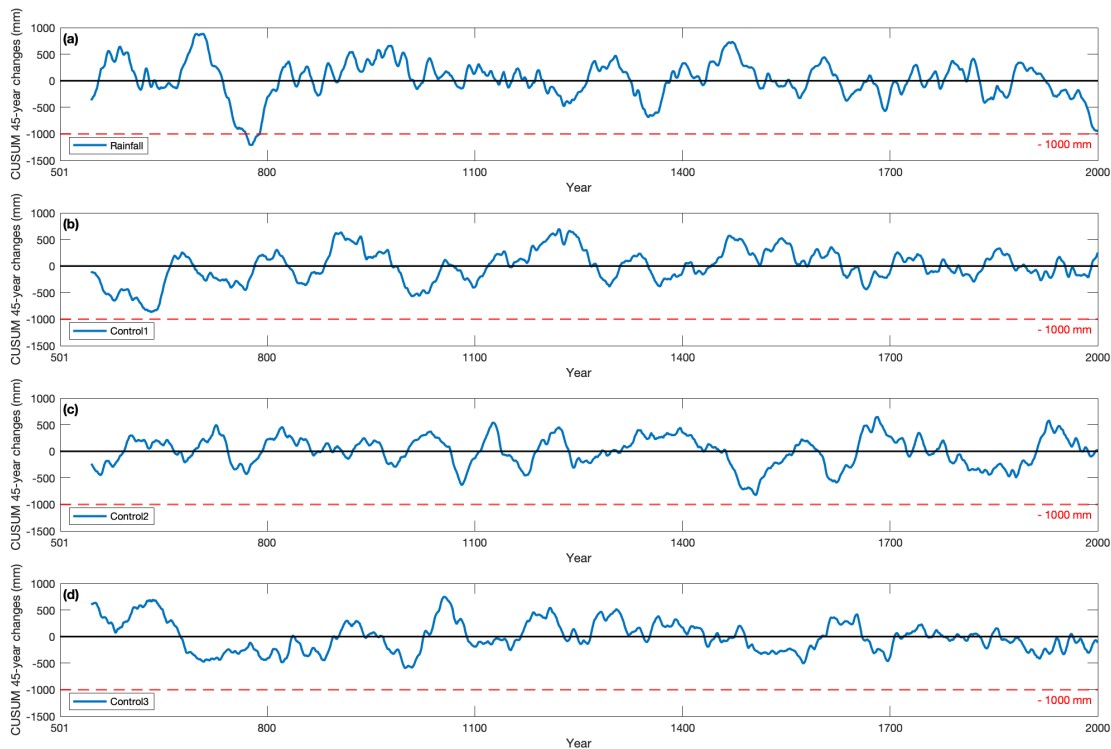

**Figure E3.** The CUSUM 45-year running change from 501 CE to 2000 CE. For each panel: (a) is the rainfall reconstruction, (b), (c) and (d) are each member of CONTROL simulations. The black horizontal line equals zero. Red horizontal dash line equals the -1000 mm (accumulated change).

*Author contributions.* YZ, SP, LJ and JR conceived the study. YZ performed the analysis. All authors contributed to the writing of the manuscript.

*Competing interests.* The authors declare that there are no competing interests.

*Acknowledgements.* This work was supported by (i) the Australian Research Council's Special Research Initiative for the Antarctic Gateway
Partnership (Project ID SR140300001), (ii) the Centre for Southern Hemisphere Oceans Research, a joint research centre between QNLM and CSIRO, and (iii) the Australia's Antarctic program (AAS 4061, 4062 and 4537).



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
