# Peer review of "Extending and understanding the South West Western Australian rainfall record using a snowfall reconstruction from Law Dome, East Antarctica"

_Climate of the Past, 2020_

## Referee Comment (RC1) · Anonymous Referee #1 · 22 Oct 2020

The authors propose a reconstruction of precipitation in South West Western Australia (SWWA) based on snowfall estimate from an ice core in East Antarctica. The authors also compare this reconstruction with model results to identify the role of various forcings and natural variability in the rainfall changes during the past 2000 years. This allows them to conclude that the recent decrease in precipitation is likely due to anthropogenic forcing. The topic of the study is very interesting. The proposed statistical analyses are well described and justified. Precipitation reconstructions are rare in the region so a long time series as the one proposed in the manuscript is more

than welcome and the analyses are useful to understand the potential origin of the reconstructed changes. However, I see several issues that need to be addressed, to reach stronger and more robust conclusions, before the paper can be accepted for publication.

General points

1. A previous study (van Ommen and Morgan, 2010) has proposed a link between precipitation in SWWA and snowfall at Law Dome. The authors confirm this link based on a least square linear regression. They discuss the validity of this statistical link but not the mechanisms that could explain it. A few lines are available in the introduction, based on the results of van Ommen and Morgan, (2010) but we have now 10 more years of data and it would be instructive to see if this confirms the conclusions reached 10 years ago or not. A point that seems puzzling is the link between an annual mean record (Law Dome) and precipitation during the growing season in SWWA. This should be discussed and justified.

2. For the reconstruction, the authors assume the stationarity of the link between snow accumulation at Law Dome and precipitation in SWWA, as classically done. However, that stationarity needs to be tested. A classical test is to separate the time series in a calibration period, over which the statistical model is developed, and a validation period, over which its skill is evaluated. In this framework, it is important to test how the magnitude of the correlation is influenced by the post 1971 trend, calibrating for instance the model over the period 1900-1971 and evaluating then the correlation over 1971-2015. This is important for the reconstruction as, if the link is weaker before 1971 (because of a strong role of the anthropogenic forcing on the observed link), this could decrease the validity of reconstruction for the pre-industrial period. In this framework, this would be nice on Figure 2 to have also the observed time series of precipitation in SWWA and snow accumulation at Law Dome for a first visual comparison of the records.

3. For the stationarity, this should also be tested using the model results to check if the model reproduce well the observed correlation for the 20th century and if the value remains stable for the previous centuries.

4. The authors compare the pre-industrial period (before 1850) with the recent past but the justification of this date is based on a commonly admitted value, not on the reconstruction itself. From a visual interpretation of the curve, it seems to me that not shift is present at that time in the series. This could be tested with the methods used to detect the shift in 1971. The differences between 22BCE-1849 and 1851-2015, shown for instance on figure 4, may be only due to the period 1971-2015. This could be tested by checking if there is also a difference between the periods 22BCE-1849 and 1851-1971. If there were no difference, this would mean that changes really occurs after 1971, not after 1851 (as discussed for Figure 6). This would also have an impact for the interpretation and for the model-data comparison as the the shift in 1850 seems clearer in the model (this should be tested too).

5. There is a good justification to use results from the CSIRO Mk3L climate model for the past millennium as not many ensemble exists with different forcing. Nevertheless, a strong point in the discussion is the post 1971 shift. The model underestimate this shift. The authors rightly argue that this may be because the model does include stratospheric ozone forcing (lines 335-336). This can be easily tested using climate model results available for this period, but including ozone forcing. A main conclusion of the study is that anthropogenic greenhouse gases have contributed to the observed decline in precipitation but this is based on one model, which do not include one of the potentially dominant forcing to explain the changes. Comparing the reconstruction with other models results, in particular those including stratospheric ozone forcing, would allow reaching stronger and more valid conclusions. It would also allow discussing the potential reasons why the shift occurs in 1971 and not earlier.

6. The way internal variability is addressed in not always adequate. The reconstruction should always be compared to an individual member (or the range of the members)

not to ensemble means. Ensemble means are useful to identify the forced response. However, ensemble means should not be used for the control run (lines 304), as for Figure 7 (if I understand well the figure). The interest of the control run is to estimate the range of internal variability as simulated by the model and this is strongly reduced for ensemble means by construction. If you use the ensemble mean, you cannot reach any conclusion on the role of internal variability in the sentence 'Examining each of the ensembles in turn, we see that CONTROL does not capture the key features of the the rainfall' (lines 307-308, two time 'the' before 'rainfall'.).

7. The comparison between the recent drying and past dry periods is too short (see for instance lines 289-291) to reach strong conclusions (lines 369-371) This is an interesting point and the way this comparison is performed should be explained and justified much more extensively.

Specific points

1. Line 19. Just by curiosity, the growing season in the region is winter (May to October)? Is it the growing season also for wheat that is mentioned line 23?

2. Line 97. What is the method used to obtain a reconstruction at a resolution of $0.05°$? This should be mentioned as this could explain part of the spatial structure obtained in Figure 1 for instance.

3. Line 136. This is probably not very important for your results but a running average is not a very good method for smoothing and other filters, simple to implement but with much better properties, are available.

4. Line 208. I do not understand why the comparison is between ensemble means and the reconstruction and which estimation of uncertainty is expected (see main point 6).

5. Line 212 'lalso' instead of 'also

6. Line 222. I do not follow 'has shown the feasibility of investigating'. Please reformulate.

7. Line 238-239. This role of recharging events seems very speculative as it is presented. Is there references to support this hypothesis?

8. The way the variable is presented on Figure 3C seems a bit strange to me. If CUSUM gives the cumulative sum of the record, what is the interest of 'the 45-year running change in the rainfall reconstruction CUSUM series' compared to a running mean on the series itself? There are maybe differences (in the magnitude in particular) but, if this diagnostics does not bring strong new information, it is probably simpler to just show the smoothed reconstruction.

9. Lines 244-245. I did not understood what is the exact difference between 'the step changes' and 'changes of the gradient' in the present framework.

10. Lines 251 'subsampled from the 501 CE to 2000 CE series'. Do you mean that you only take a part of the series?

11. Caption of Figure 5. What is 'the reconstruction in period 1850–2000 CE from the 501–2000 CE series', here too a part of the series?

12. Table 5. What means 'Sample represents to each rainfall simulations'?

13. For me, all the appendices could be moved to a supplementary material.

.

---

## Referee Comment (RC2) · Matthew H. England (Referee) · 26 Jan 2021

Review of the paper: "Extending and understanding the southwest Western Australian rainfall record using a snowfall reconstruction from Law Dome, East Antarctica"
By: Zheng, Jong, Phipps, Roberts, Moy, Curran, and van Ommen
Submitted to *Climate of the Past*

This paper extends a previous estimation of Southwest Western Australian (SWWA) rainfall using a snowfall reconstruction from Law Dome, East Antarctica. Overall the paper is well written and thorough, with appropriate use of statistical methods and suitable conclusions given the analyses obtained. It's a nice contribution to the field, and I very much enjoyed reading the paper.

Nonetheless, I have some suggestions that the authors should take into consideration when preparing a revised manuscript. Once suitably revised, I expect to recommend the paper for publication in *Climate of the Past*. I am happy to look at the revised manuscript when it's available for review.

General comments:

1. The square of the correlation coefficients ($R^2$) reveals the fraction of variance in SWWA rainfall explained by the snowfall reconstruction at Law Dome. So this explained variance maxes out at only around ~25-35% (i.e. 0.5 - 0.6 squared). I think this needs to be acknowledged in the paper (for example, added as a new column in Table 3). This relatively low explained variance suggests that tropical and subtropical influences also play a significant role in driving rainfall variations over the region. Implying that the rainfall estimates from the Law Dome snow proxies carry considerable uncertainty. It's important to expand on this point in the discussion / conclusions.

2. Related to this point, there are several studies that are not yet cited in the paper that make this link from the tropics and subtropics to SWWA rainfall, including (but not limited to) the following:

England, M. H., C. C. Ummenhofer, and A. Santoso, 2006: Interannual rainfall extremes over southwest Western Australia linked to Indian Ocean climate variability. *J. Climate*, **19**, 1948-1969.

Ummenhofer, C. C., A. Sen Gupta, M. J. Pook, and M. H. England, 2008: Anomalous rainfall over southwest Western Australia forced by Indian Ocean sea surface temperatures, *J. Climate*, **21**, 5113-5134.

Smith, I. N., P. McIntosh, T. J. Ansell, C. J. C. Reason, and K. McInnes, 2000: Southwest Western Australian winter rainfall and its association with Indian Ocean climate variability. *Int. J. Climatol.*, **20**, 1913–1930.

The authors need to expand their discussion of the tropical and subtropical influence on SWWA rainfall, perhaps around lines 35 - 50 of the Introduction, or after that paragraph. And more clearly acknowledge that the SAM is not the sole driver of SWWA rainfall variability. Mention is made of this, but it needs to be expanded in relation to tropical and subtropical influences.

3. It's curious that the relationship between snowfall at Law Dome and SWWA rainfall is maximised with a 5–6 year low pass filter. This surprised me; I would've thought the annual signal would dominate. Is there any climatic reason for this? Dominant frequencies of variability of both the SAM and SWWA rainfall do not include a 5-year signal as far as I know. It would be good for the authors to expand on this discussion a little — the statistical analysis is clear, but what is the climatic interpretation?

4. The analysed rainfall data only extends up to 2015 in this study (lines 270-272). I realise that the snowfall accumulation data may not be available after this year, but the reference to whether the drought and prolonged dry period might be continuing after 2015 is made without being quantified here. I suggest that the authors at least clarify this for rainfall post-2015? Did the dry spell continue during 2016-2020? Otherwise the reader is left hanging here.

5. Figure 6 is very nice!

6. The authors note that the model has a dry bias (lines 284-285 and also Table E1). With SWWA rainfall impacted by the Southern Annular Mode, and with the westerly winds and the storm tracks generally having an equatorward bias in climate models, this is kind of surprising (this bias should lead to a wetter than observed SWWA region). Can the authors speculate on the possible reasons for the model's dry bias?

7.  The use of the single and multiple forcing coupled model experiments is very nice, allowing the authors to separate out various climate forcings, including solar, greenhouse gases, volcanic and orbital.  I liked this analysis and the associated discussion is very clear.  An experiment with just stratospheric ozone depletion forcing would have been interesting as well, given the SAM link, but if this is not available to include, no problem.

8.  The figures are generally of good quality, but the font sizes of all axis labels and figure legends etc. is often far too small.  Definitely worth fixing this before publication.

9. The final sentence or two of the paper is devoted to mentioning the two (possible) previous prolonged drought events over SWWA in the proxy-estimated record.  To me this is an interesting finding, but not the major "take home" finding of the study. I suggest the authors add a final sentence beginning "However,…" wherein they describe the finding that GHG forcing is the likely driver of the SWWA rainfall decline since the early 1970s.

10.  Figure B1:  There are peaks in both panels (a) and (b) at two years. What is the climatic interpretation of this?

Minor suggestions:

11. Lines 43-44:  Thompson and Solomon 2002 do not show analyses of the Southern Annular Mode link to SWWA rainfall. This reference should be changed to Thompson et al. 2011:

Thompson, D. W. J., S. Solomon, P. J. Kushner, M. H. England, K. M. Grise and D. J. Karoly, 2011: Signatures of the Antarctic ozone hole in Southern Hemisphere surface climate change, *Nature Geoscience*, **4**, 741-749.

12.  Line 58:  Add a citation to Goodwin et al. 2004 here alongside the reference to van Ommen and Morgan 2010:

Goodwin, I. D., , T. D. van Ommen, , M. A. J. Curran, , and P. A. Mayewski, 2004: Mid latitude winter climate variability in the South Indian and southwest Pacific regions since 1300 ad. *Climate Dyn.*, **22**, 783–794.

13.  Lines 104-105:  England et al. (2006) also analyse the quality of the Mk3L model simulations in relation to observed interannual variability of SWWA rainfall.  Perhaps cite their findings here.

14.  I found the shorthand term "MASK" a little obtuse for the region of statistical significance of the rainfall signal over SWWA.  I think it is okay to identify that region upfront as the authors have done, but then just say that hereafter, when describing SWWA rainfall, it is taken to indicate the region delineated by this area in Fig. 1.  I think the resulting text will be clearer that way.

15.  The amount of explained variance $R^2$ should be added to Table 3, expressed as a percentage.  I think this would add to the information provided in that table.

16.  Lines 169-170: are there also non-significant stations within this MASK region?   If so, the authors should point this out.

17.  Typo, Figure 2 caption, line 1:   plor -> plot

18.  The term "model outputs" is used often in the paper, I would suggest changing this to terms like "the model simulations", "the model runs", "the model experiments" (etc.) throughout the manuscript.

19.  Figure E1: I understand why the authors may wish to include this diagram for completeness, but each of the panels looks basically the same as each of the other panels.  So I wonder if there is any utility in this diagram being included?

Matthew England
UNSW

---------- END OF REVIEW ----------

---

## Editor Comment (EC1) · Eric Wolff (Editor) · 27 Jan 2021

I am sorry that you had to wait longer than we would like for the reviews to be completed. However you now have two very helpful reviews of your paper. They both see good potential for the paper to be published eventually but they make some fairly major points about issues such as stationary of the link between the Antarctic and Australian signals, and about the non-polar influences. Please address each issue raised thoroughly in your responses to the reviews.

---

## Author Comment (AC1) · 5 Mar 2021

**Extending and understanding the South West Western Australian rainfall record using a snowfall reconstruction from Law Dome, East Antarctica**

by Yaowen Zheng, Lenneke M. Jong, Steven J. Phipps, Jason L. Roberts, Andrew D. Moy, Mark A. J. Curran, and Tas D. van Ommen,

submitted to Climate of the Past (https://doi.org/10.5194/cp-2020-124)

We are glad that the reviewer thinks the topic of the manuscript is very interesting. We thank them for the time they have spent reading and reviewing it. We respond to each of the general and specific points below. The reviewer's comments are shown in **bold text**, replies are shown in normal text, text from the original manuscript is shown in blue, and proposed changes
5    to the manuscript are shown in red.
* * *
**The authors propose a reconstruction of precipitation in South West Western Australia (SWWA) based on snowfall estimate from an ice core in East Antarctica. The authors also compare this reconstruction with model results to identify the role of various forcings and natural variability in the rainfall changes during the past 2000 years. This**
10   **allows them to conclude that the recent decrease in precipitation is likely due to anthropogenic forcing. The topic of the study is very interesting. The proposed statistical analyses are well described and justified. Precipitation reconstructions are rare in the region so a long time series as the one proposed in the manuscript is more than welcome and the analyses are useful to understand the potential origin of the reconstructed changes. However, I see several issues that need to be addressed, to reach stronger and more robust conclusions, before the paper can be**
15   **accepted for publication.**

**General points**

**1** **A previous study (van Ommen and Morgan, 2010) has proposed a link between precipitation in SWWA and snowfall at Law Dome. The authors confirm this link based on a least square linear regression. They discuss the validity of this statistical link but not the mechanisms that could explain it. A few lines are available in the**
20 **introduction, based on the results of van Ommen and Morgan (2010) but we have now 10 more years of data and it would be instructive to see if this confirms the conclusions reached 10 years ago or not. A point that seems puzzling is the link between an annual mean record (Law Dome) and precipitation during the growing season in SWWA. This should be discussed and justified.**

We thank the reviewer for this comment, however, this paper does not explore the mechanisms for the link itself. These were
25 considered in van Ommen and Morgan (2010), which points to variations in meridional circulation south of Australia. We have added one sentence at around Line 61 to explicitly mention this mechanism of van Ommen and Morgan (2010) in the text. Line 61: "The relationship consists of a statistically significant negative correlation between winter JJA mean SWWA regional rainfall and Law Dome, East Antarctica, and the mechanism of this link is the variations in meridional circulation south of Australia(van Ommen and Morgan, 2010)."
30 The Law Dome ice core record has a well-defined annual chronology that allows annual snowfall to be determined, but a suitable method for extracting sub-annual accumulation from the ice core has not been developed. While an accurately registered snowfall accumulation series for winter would be desirable and might be expected to correlate more strongly with SWWA rainfall, such seasonal accumulation data are not available.

**2** **For the reconstruction, the authors assume the stationarity of the link between snow accumulation at Law Dome**
35 **and precipitation in SWWA, as classically done. However, that stationarity needs to be tested. A classical test is to separate the time series in a calibration period, over which the statistical model is developed, and a validation period, over which its skill is evaluated. In this framework, it is important to test how the magnitude of the correlation is influenced by the post 1971 trend, calibrating for instance the model over the period 1900-1971 and evaluating then the correlation over 1971-2015. This is important for the reconstruction as, if the link is weaker**
40 **before 1971 (because of a strong role of the anthropogenic forcing on the observed link), this could decrease the validity of reconstruction for the pre-industrial period. In this framework, this would be nice on Figure 2 to have also the observed time series of precipitation in SWWA and snow accumulation at Law Dome for a first visual comparison of the records.**

Thank you for pointing this out. However, we think this might be a validation issue instead of a stationarity one. To validate
45 the reconstruction, we have now performed a jackknife analysis over the period 1900-2015 for the mean AWAP data in the "MASK" region. Specifically:

1. We determine the autocorrelation length by calculating the autocorrelation at various time lags (e.g. lag-1, lag-2, lag-3, ..., lag-6, ...) on the unsmoothed observational data until the autocorrelation coefficient decreases to zero.

2. We use this autocorrelation length to perform a (modified) jackknife analysis.

50

3. For each subsample, we perform 6-year smoothing then calculate the correlation coefficient and construct the linear model.

The autocorrelation time-scale is 12 years. The results of the jackknife analysis are shown in Table 1. The correlation coefficients for the individual members of the jackknife ensemble are tightly clustered around the value of -0.597 for the full reconstruction. Furthermore, the overlap between the 95% confidence intervals for the gradient and intercept indicates that

55 the models derived for each jackknife ensemble member are statistically indistinguishable at the 5% probability level. This demonstrates the robustness of the reconstruction technique.

**Table 1.** The correlation coefficient between the smoothed ice core record and the mean of the AWAP rainfall in the "MASK" region. x1 and Intercept are the coefficients for the linear model: Rain = Snow * ( x1 $\pm$ 95% CI ) + Intercept $\pm$ 95% CI mm/year. The 95% confidence interval (CI) is estimated by multiplying the standard error of the model by 1.96.

| Period | Correlation coefficient | x1 | Intercept (mm/year) |
|---|---|---|---|
| 1912–2015 | -0.536 | -0.385 $\pm$ 0.118 | 667 $\pm$ 85 |
| 1900–1911 & 1924–2015 | -0.513 | -0.363 $\pm$ 0.118 | 651 $\pm$ 85 |
| 1900–1923 & 1936–2015 | -0.532 | -0.378 $\pm$ 0.117 | 662 $\pm$ 85 |
| 1900–1935 & 1948–2015 | -0.524 | -0.405 $\pm$ 0.128 | 684 $\pm$ 93 |
| 1900–1947 & 1960–2015 | -0.545 | -0.407 $\pm$ 0.122 | 685 $\pm$ 88 |
| 1900–1959 & 1972–2015 | -0.499 | -0.385 $\pm$ 0.130 | 671 $\pm$ 94 |
| 1900–1971 & 1984–2015 | -0.506 | -0.427 $\pm$ 0.141 | 699 $\pm$ 101 |
| 1900–1983 & 1996–2015 | -0.492 | -0.374 $\pm$ 0.129 | 662 $\pm$ 92 |
| 1900–1995 & 2008–2015 | -0.512 | -0.356 $\pm$ 0.116 | 651 $\pm$ 82 |
| 1900–2007 | -0.593 | -0.379 $\pm$ 0.098 | 670 $\pm$ 71 |
| 1900–2015 | -0.597 | -0.389 $\pm$ 0.114 | 672 $\pm$ 82 |

**3  For the stationarity, this should also be tested using the model results to check if the model reproduce well the observed correlation for the 20th century and if the value remains stable for the previous centuries.**

The ability of the Law Dome accumulation record to be used to make inferences about shifts in SWWA rainfall hinges upon

60 the stability of the relationship between accumulation at Law Dome and the Southern Hemisphere atmospheric circulation. To explore this, we use the three members of ensemble OGSV to calculate the correlation between precipitation at Law Dome and

Southern Hemisphere mean sea level pressure. The results are shown in the following Figure 1, for the periods 1975-2000 CE (top row), 1851-2000 CE (middle row) and 501-2000 CE (bottom row).

[Figure]

**Figure 1.** Correlation between precipitation at Law Dome and Southern Hemisphere mean sea level pressure for each member of OGSV.

Interannual variability gives rise to some differences between the individual model siulations, particular on shorter timescales.

65 However, the pattern of positive correlations over the high-latitude Southern Ocean, surrounded by negative correlations at mid-latitudes, is consistent with the observed relationship (as shown in van Ommen and Morgan, 2010) and can be seen to be stable over the full 1500 years of the climate model simulations.

**4  The authors compare the pre-industrial period (before 1850) with the recent past but the justification of this date is based on a commonly admitted value, not on the reconstruction itself. From a visual interpretation of the curve, it**

70 **seems to me that not shift is present at that time in the series. This could be tested with the methods used to detect the shift in 1971. The differences between 22 BCE-1849 and 1851-2015, shown for instance on figure 4, may be only due to the period 1971-2015. This could be tested by checking if there is also a difference between the periods 22 BCE-1849 and 1851- 1971. If there were no difference, this would mean that changes really occurs after 1971, not after 1851 (as discussed for Figure 6). This would also have an impact for the interpretation and for the**

75 **model-data comparison as the the shift in 1850 seems clearer in the model (this should be tested too).**

Thank you. The BREAKFIT analysis on the CUSUM of the reconstruction showed the break-point is at 1971. However, the BREAKFIT analysis only picks the largest change in the gradient, with a single break-point as the output. This means that we did not know if there were other significant changes in the gradient, apart from at 1971. It is therefore possible that the differences between 22 BCE–1849 and 1851–2015 are due to the period 1971–2015.

80 To test this, we independently perform the two-sample KS test on the additional combination of periods. The results are shown in Table 2.

**Table 2.** The results for independent KS tests for different time periods from the rainfall time series. The result "same" means we cannot reject the null hypothesis (that the two samples of data are from the same distribution), while "different" means we can reject the null hypothesis ($p < 0.05$).

| Sample | 501–1849 VS 1851–1970 |
| --- | --- |
| CONTROL | same |
| O | same |
| OG | different |
| OGS | same |
| OGSV | different |
| Rainfall reconstruction | different |

The model simulations suggest an anthropogenic influence beginning to emerge after 1850, which is at the limit of detectability at first because of natural climate variability. The reconstuction shows a secondary change due to anthropogenic influence during the epoch 1850-1970.

**5** **There is a good justification to use results from the CSIRO Mk3L climate model for the past millennium as not many ensemble exists with different forcing. Nevertheless, a strong point in the discussion is the post 1971 shift. The model underestimate this shift. The authors rightly argue that this may be because the model does include stratospheric ozone forcing (lines 335-336). This can be easily tested using climate model results available for this period, but including ozone forcing. A main conclusion of the study is that anthropogenic greenhouse gases have contributed to the observed decline in precipitation but this is based on one model, which do not include one of the potentially dominant forcing to explain the changes. Comparing the reconstruction with other models results, in particular those including stratospheric ozone forcing, would allow reaching stronger and more valid conclusions. It would also allow discussing the potential reasons why the shift occurs in 1971 and not earlier.**

We agree that using multiple models (especially those which include stratospheric ozone forcing) would be helpful in better understanding the rainfall shift in 1971. This is beyond the scope of the current study, but would be a very useful avenue for further investigations into the observed changes in SWWA rainfall.

We have added some text in the paragraph at Lines 364-367: "Both the reconstruction and the climate model simulations suggest that the drying trend began earlier than the 1970s. However, the model simulations do not capture the acceleration in the reconstructed drying trend at around 1971 CE. This suggests that this acceleration cannot be attributed to external forcings, or at least not any of the forcings considered in this study. Either natural variability or stratospheric ozone depletion are potential alternative explanations (e.g. Cai and Shi, 2005). A investigation into possible ozone forcing and a comparison against multiple climate models is a promising avenue for future work."

**6** **The way internal variability is addressed in not always adequate. The reconstruction should always be compared to an individual member (or the range of the members) not to ensemble means. Ensemble means are useful to identify the forced response. However, ensemble means should not be used for the control run (lines 304), as for Figure 7 (if I understand well the figure). The interest of the control run is to estimate the range of internal variability as simulated by the model and this is strongly reduced for ensemble means by construction. If you use the ensemble mean, you cannot reach any conclusion on the role of internal variability in the sentence 'Examining each of the ensembles in turn, we see that CONTROL does not capture the key features of the the rainfall' (lines 307-308, two time 'the' before 'rainfall'.).**

Thank you, but we do not fully agree. In general, the ensemble mean has a better signal-to-noise ratio and is therefore more representative of the response to external forcings than any individual ensemble member. For this reason, the bulk of the analysis is performed on ensemble means. We intentionally processed the CONTROL simulations in the same way as the forced simulations to derive independent estimates of the amplitude of noise in the four forced ensembles.

Where we are considering internal variability, for example at Lines 286–288, we examine the individual ensemble members separately.

Thank you for pointing out this typographical error. We will make the change in Line 307–308: "Examining each of the ensembles in turn, we see that CONTROL does not capture the key features of  the rainfall reconstruction (Figure 7)."

**7   The comparison between the recent drying and past dry periods is too short (see for instance lines 289-291) to reach strong conclusions (lines 369-371) This is an interesting point and the way this comparison is performed should be explained and justified much more extensively.**

We agree and we have revised the manuscript accordingly. Following "...consistent with the findings of Cai and Shi (2005)." at Line 288, we have replaced the text with the following paragraph:

"A number of prolonged droughts that approach the magnitude of the current integrated rainfall deficit are apparent in the reconstruction (Figure 3c), as discussed in Section 4.1. These prior drought epochs occurred during the pre-industrial era, suggesting that they may have arisen through natural climate variability or natural forcings such as volcanoes. To explore this within the model simulations, the 45-year changes in the CUSUM for the pre-industrial control simulation are shown in Figure E3. A number of prolonged droughts are also apparent in all three ensemble members, for example at years 589–633 CE in Control1 (Figure E3b). This supports the hypothesis that prolonged droughts of a similar magnitude to the currently observed drought can arise to natural climate variability or natural forcings."

At Line 369, we have replaced "Droughts of similar duration and magnitude occur in an unforced pre-industrial control simulation, suggesting that these events may have arisen through natural climate variability." with " Droughts of similar duration and magnitude also occur in an unforced pre-industrial control simulation. The model simulations therefore support the hypothesis that these pre-industrial drought events may have arisen through natural climate variability. However, forced climate model simulations indicate that anthropogenic greenhouse gases are the dominant driver of the rainfall reduction in SWWA since the early 1970s."

**Specific points**

**8   Line 19. Just by curiosity, the growing season in the region is winter (May to October)? Is it the growing season also for wheat that is mentioned line 23?**

Rainfall in SWWA is seasonal with ~75% of the annual total falling in May-October (Ludwig and Asseng, 2006). This corresponds to the growing season for wheat cropping in this region.

**9   Line 97. What is the method used to obtain a reconstruction at a resolution of 0.05°? This should be mentioned as this could explain part of the spatial structure obtained in Figure 1 for instance.**

The AWAP data is 0.05° resolution gridded data. Jones et al. (2009) describes the method used to generate the gridded dataset. We describe the method that we use to generate the reconstruction at Line 152 to Line 157, "To define a region with statistically

significant correlation between observed (AWAP) and reconstructed (DSS) rainfall over SWWA, we independently calculate the correlation coefficient and test its significance for 110 Local Government Areas (Australian Goverment, 2020) (details in Appendix C) in WA. There are 9 Local Government Areas smaller than the 0.05° longitude * 0.05° latitude geospatial resolution of the AWAP data grid. Therefore, we actually calculate and test 109 (Sorry, this should be "101" instead of "109") areas. There are 52 areas that are statistically significant (6-year window, p < 0.05). We combine these 52 areas to define the significant region (Figure 1) over SWWA. For convenience, we name this significant region as "MASK"."We took out that 9 areas but looking at the rest of 101 areas. The Australian Government (2020) data we used for every single 101 areas has the same 0.05° resolution and grids as the AWAP data. For each area, there is a mask matrix– the value equals "1" within the area and "0" outside the area. So we did Dot Product between the AWAP rainfall data and every single LGA to generate the AWAP rainfall in each area and independently calculated the correlation coefficient and tested its significance between each of 101 areas' AWAP rainfall and snowfall record, then we did addition on 52 significant LGAs' mask matrices to generate the mask matrix of the region "MASK". Then, we did Dot Product between the AWAP rainfall data and the "MASK" mask matrix to generate the AWAP rainfall in "MASK" region and calculated the mean "MASK" region rainfall which was used to obtain a reconstruction at a resolution of 0.05°.

We thank you for pointing out the spatial structure obtained in Figure 1. We apologize for this. We think the spatial structure (e.g. some gaps between the green line and the map) was due to the difference between the Western Australia land mask and the LGA masks. We preformed a scalar multiplication between the AWAP rainfall and the Western Australia land mask before performing the calculations to get remove the rainfall data in ocean areas, increase the speed of calculations and reduce memory usage.

**10   Line 136. This is probably not very important for your results but a running average is not a very good method for smoothing and other filters, simple to implement but with much better properties, are available.**

We thank the reviewer for this helpful suggestion. We will take it into consideration for future work.

**11   Line 208. I do not understand why the comparison is between ensemble means and the reconstruction and which estimation of uncertainty is expected (see main point 6).**

We have addressed this in our reply to main point 6 above.

**12   Line 212 'lalso' instead of 'also**

Thank you for carefully pointing out this typographical error. We have changed in Line 212: "We also estimate the adapted degrees of freedom (DF) using the Welch–Satterthwaite equation:"

**13   Line 222. I do not follow 'has shown the feasibility of investigating'. Please reformulate.**

We agree this wording is unclear. We have changed in Line 222 from "This rainfall reconstruction has shown the feasibility of investigating the longer-term context of rainfall variability over SWWA before the instrument-era." to "This rainfall reconstruction has shown that it is possible to use the ice core to investigate the longer-term context of rainfall variability over SWWA before the instrument-era."

**14   Line 238-239. This role of recharging events seems very speculative as it is presented. Is there references to support this hypothesis?**

Yes, there is a reference to support this hypothesis. Gallant et al. (2013) states "Soil moisture is regulated by multiple hydroclimatic processes, such as evaporation, run-off and groundwater recharge, which are potentially important drivers of drought variability."

We now cite Gallant et al. (2013) at Lines 238-239: "...and this lack of recharging events might have more of an impact than the shift in the mean(e.g., Gallant et al., 2013) ."

**15   The way the variable is presented on Figure 3C seems a bit strange to me. If CUSUM gives the cumulative sum of the record, what is the interest of 'the 45-year running change in the rainfall reconstruction CUSUM series' compared to a running mean on the series itself? There are maybe differences (in the magnitude in particular) but, if this diagnostics does not bring strong new information, it is probably simpler to just show the smoothed reconstruction.**

Thank you for pointing this out, but we do not fully agree. We think the 45-year running change in the rainfall reconstruction CUSUM series does bring important new information. As we state at Lines 272-276, we have found there was a prolonged drought in SWWA from 1971 CE to 2015 CE that might be ongoing. We calculated the 45-year running change in the rainfall reconstruction CUSUM series to "highlight dry epochs of an equivalent duration to the observed drought to date".Using this analysis, We found the important new information that there are "two comparable prior epochs during the past two millennia, lasting from around 385–429 CE and 732–776 CE (Figure 3c)" with similar duration and intensity.

**16   Lines 244-245. I did not understood what is the exact difference between 'the step changes' and 'changes of the gradient' in the present framework.**

Thank you for pointing this out. We apologize for this wording. We have changed at Lines 243-245: "In order to accurately determine the changes in rainfall after 1850 CE, we calculate the CUSUM time series (Equation 3) for rainfall time series (Figure 3a)  and use BREAKFIT analysis  to identify any significant changes of the gradient."

**17    Lines 251 'subsampled from the 501 CE to 2000 CE series'. Do you mean that you only take a part of the series?**

205    Thank you for pointing this out. We agree that this text is unclear. We have changed at Line 251: "Figure 5 shows the rainfall reconstruction CUSUM time series from 1850 CE to 2000 CE  subsampled from the 501 CE to 2000 CE series."

**18    Caption of Figure 5. What is 'the reconstruction in period 1850–2000 CE from the 501–2000 CE series', here too a part of the series?**

Thank you for pointing this out. We agree that this text is unclear. We have changed the caption of Figure 5: "CUSUM time
210    series on rainfall reconstruction in period 1850–2000 CE  from the 501–2000 CE series."

**19    Table 5. What means 'Sample represents to each rainfall simulations'?**

Thank you for pointing this out. We agree that this text is unclear.

We have changed the caption of Table 5: "The RMSE and difference in slope calculated between the rainfall reconstruction and simulations in the period 1972–2000 CE. Sample represents to each rainfall simulations. The difference in slope is the
215    difference of the slope of the Ordinary Least Square linear fitted CUSUM time series data for each sample to the slope of the fitted CUSUM time series data for rainfall reconstruction in the period 1972–2000 CE."

**20    For me, all the appendices could be moved to a supplementary material.**

Thank you for this suggestion. We agree and will move all the appendices to supplementary material.

**References**

220    Australian Government: Local Government Area, https://data.gov.au/data/dataset/8a8e0037-e474-422f-8026-241c7c88551f, 2020.

Gallant, A. J., Reeder, M. J., Risbey, J. S., and Hennessy, K. J.: The characteristics of seasonal-scale droughts in Australia, 1911–2009, International Journal of Climatology, 33, 1658–1672, 2013.

Jones, D. A., Wang, W., and Fawcett, R.: High-quality spatial climate data-sets for Australia, Australian Meteorological and Oceanographic Journal, 58, 233, 2009.

225    Ludwig, F. and Asseng, S.: Climate change impacts on wheat production in a Mediterranean environment in Western Australia, Agricultural Systems, 90, 159–179, 2006.

van Ommen, T. D. and Morgan, V.: Snowfall increase in coastal East Antarctica linked with southwest Western Australian drought, Nature Geoscience, 3, 267–272, 2010.

---

## Author Comment (AC2) · 5 Mar 2021

**Extending and understanding the South West Western Australian rainfall record using a snowfall reconstruction from Law Dome, East Antarctica**

by Yaowen Zheng, Lenneke M. Jong, Steven J. Phipps, Jason L. Roberts, Andrew D. Moy, Mark A. J. Curran, and Tas D. van Ommen,

submitted to Climate of the Past (https://doi.org/10.5194/cp-2020-124)

We are glad that the reviewer enjoyed reading the manuscript. We thank him for the time that he spent reading and reviewing it. We respond to each of the general and specific points below. The reviewer's comments are shown in **bold text**, replies are shown in normal text, text from the original manuscript is shown in blue, and proposed changes to the manuscript are shown

5    in red.
* * *
**This paper extends a previous estimation of Southwest Western Australian (SWWA) rainfall using a snowfall reconstruction from Law Dome, East Antarctica. Overall the paper is well written and thorough, with appropriate use of statistical methods and suitable conclusions given the analyses obtained. It's a nice contribution to the field, and I**

10   **very much enjoyed reading the paper.**

**Nonetheless, I have some suggestions that the authors should take into consideration when preparing a revised manuscript. Once suitably revised, I expect to recommend the paper for publication in Climate of the Past. I am happy to look at the revised manuscript when it's available for review.**

**General comments:**

**1  The square of the correlation coefficients (R2) reveals the fraction of variance in SWWA rainfall explained by the snowfall reconstruction at Law Dome. So this explained variance maxes out at only around 25-35% (i.e. 0.5 - 0.6 squared). I think this needs to be acknowledged in the paper (for example, added as a new column in Table 3). This relatively low explained variance suggests that tropical and subtropical influences also play a significant role in driving rainfall variations over the region. Implying that the rainfall estimates from the Law Dome snow proxies carry considerable uncertainty. It's important to expand on this point in the discussion / conclusions.**

We thank the reviewer for this comment. We agree that the explained variance of ~25-35% should be acknowledged in the manuscript and we have revised the text accordingly at Lines 169-171: "...showing the consistency with the significance of MASK. The square of the correlation coefficients have shown the explained variance are maximum at around 30-40%. The tropics and subtropics can play an important role in driving rainfall changes in SWWA (Smith et al., 2000; England et al., 2006; Ummenhofer et al., 2008). Using the Law Dome ice core snow accumulation proxy to reconstruct the SWWA rainfall has non-negligible uncertainty. However, in general the proxies are rare in such area suggesting this 30-40% makes a valuable contribution to our ability to reconstruct past climate. Therefore,..."

**2  Related to this point, there are several studies that are not yet cited in the paper that make this link from the tropics and subtropics to SWWA rainfall, including (but not limited to) the following:**

**England, M. H., C. C. Ummenhofer, and A. Santoso, 2006: Interannual rainfall extremes over southwest Western Australia linked to Indian Ocean climate variability. J. Climate, 19, 1948-1969.**

**Ummenhofer, C. C., A. Sen Gupta, M. J. Pook, and M. H. England, 2008: Anomalous rainfall over southwest Western Australia forced by Indian Ocean sea surface temperatures, J. Climate, 21, 5113- 5134.**

**Smith, I. N., P. McIntosh, T. J. Ansell, C. J. C. Reason, and K. McInnes, 2000: Southwest Western Australian winter rainfall and its association with Indian Ocean climate variability. Int. J. Climatol., 20, 1913–1930.**

**The authors need to expand their discussion of the tropical and subtropical influence on SWWA rainfall, perhaps around lines 35 - 50 of the Introduction, or after that paragraph. And more clearly acknowledge that the SAM is not the sole driver of SWWA rainfall variability. Mention is made of this, but it needs to be expanded in relation to tropical and subtropical influences.**

We thank the reviewer for these suggestions and have added these references to the manuscript.

We have also expanded the discussion of the tropical and subtropical influences on SWWA rainfall from Line 47, as follows:

This shift, in conjunction with the increase in anthropogenic greenhouse gases over this period, may be responsible for at least part of the reduction in SWWA rainfall (Cai and Shi, 2005).

Negative sea surface temperature anomalies in the eastern Indian Ocean and positive sea surface temperature anomalies in the central subtropical Indian Ocean are related to dry years in SWWA (Ummenhofer et al., 2008). Some indication was found that mean sea level pressure anomalies over the Indian Ocean drive Indian Ocean sea surface temperature anomalies and SWWA rainfall, but this link did not appear robust at the interannual time scale (Smith et al., 2000). A link between SWWA rainfall extremes and large-scale Indian Ocean climate was found due to moisture advection onto the SWWA coast (England et al., 2006). This is subject to influence from the large-scale wind field over the eastern and southeastern Indian Ocean, which may contribute to SWWA rainfall extremes (England et al., 2006). This link between the Indian Ocean and SWWA rainfall may be influenced by the IOD–ENSO link (England et al., 2006). Thus, the tropical and subtropical Indian and Pacific Oceans are both likely to play a role in SWWA rainfall variations, but not the only role. Changes of a similar magnitude to those observed can potentially also arise through natural multidecadal climate variability (Cai and Shi, 2005). Thus the drivers of the winter rainfall attenuation in SWWA are still unclear.

**3   It's curious that the relationship between snowfall at Law Dome and SWWA rainfall is maximised with a 5–6 year low pass filter. This surprised me; I would've thought the annual signal would dominate. Is there any climatic reason for this? Dominant frequencies of variability of both the SAM and SWWA rainfall do not include a 5-year signal as far as I know. It would be good for the authors to expand on this discussion a little — the statistical analysis is clear, but what is the climatic interpretation?**

Thank you for this comment. We used the same methodology as van Ommen and Morgan (2010), who found that smoothing was required due to noise introduced by surface processes at the ice core site. We have revised the text accordingly at Lines 134-136: "Low-pass filtering (or smoothing) the data increases the correlation of the precipitation time-series between SWWA and Law Dome (van Ommen and Morgan, 2010). Annual-scale noise arises from site surface processes and snow accumulation variability, and is ameliorated by the smoothing."

**4** **The analysed rainfall data only extends up to 2015 in this study (lines 270-272). I realise that the snowfall accumulation data may not be available after this year, but the reference to whether the drought and prolonged dry period might be continuing after 2015 is made without being quantified here. I suggest that the authors at least clarify this for rainfall post-2015? Did the dry spell continue during 2016-2020? Otherwise the reader is left hanging here.**

Thank you for this comment. We have downloaded and plotted (Figure 1) the latest rainfall data from the Australian Government Bureau of Meteorology for the growing season in southwestern Australia from 1900 to 2020. The data is available at: http://www.bom.gov.au/climate/change

The mean growing season rainfall in southwestern Australia from 1900 to 1970 is 564.97 mm, from 1972 to 2014 is 481.26 mm and from 2016 to 2020 is 431.35 mm. The drought continued during 2016–2020.

[Figure]

**Figure 1.** Time series of BoM rainfall in southwestern Australia at growing season from 1900 to 2020.

**5** **Figure 6 is very nice!**

Thank you!

**6** **The authors note that the model has a dry bias (lines 284-285 and also Table E1). With SWWA rainfall impacted by the Southern Annular Mode, and with the westerly winds and the storm tracks generally having an equatorward bias in climate models, this is kind of surprising (this bias should lead to a wetter than observed SWWA region). Can the authors speculate on the possible reasons for the model's dry bias?**

The model has a relatively coarse spatial resolution: for example, the SWWA region is represented by a single point on the atmosphere model grid. The model is not therefore able to capture finer-scale topographic variations, with the consequence

that precipitation can tend to be too low along the western margins of continental landmasses. We speculate that this accounts for the dry bias that we find in the current study.

**7   The use of the single and multiple forcing coupled model experiments is very nice, allowing the authors to separate out various climate forcings, including solar, greenhouse gases, volcanic and orbital. I liked this analysis and the associated discussion is very clear. An experiment with just stratospheric ozone depletion forcing would have been interesting as well, given the SAM link, but if this is not available to include, no problem.**

We thank the reviewer for this comment.

In regard to ozone depletion, the CSIRO Mk3L climate system model does not include atmospheric chemistry and so no simulations exist that include changes in stratospheric ozone. While we acknowledge that we could include additional models, such as the CMIP5 or CMIP6 ensembles, we feel that this is beyond the scope of the current study.

**8   The figures are generally of good quality, but the font sizes of all axis labels and figure legends etc. is often far too small. Definitely worth fixing this before publication.**

We have modified the figures accordingly.

**9   The final sentence or two of the paper is devoted to mentioning the two (possible) previous prolonged drought events over SWWA in the proxy-estimated record. To me this is an interesting finding, but not the major "take home" finding of the study. I suggest the authors add a final sentence beginning "However,..." wherein they describe the finding that GHG forcing is the likely driver of the SWWA rainfall decline since the early 1970s.**

Thank you for your suggestion. We have accepted your suggestion and added a final sentence accordingly: "However, forced climate model simulations indicate that anthropogenic greenhouse gases are the dominant driver of the rainfall reduction in SWWA since the early 1970s."

**10   Figure B1: There are peaks in both panels (a) and (b) at two years. What is the climatic interpretation of this?**

This is likely to be a combination of many different processes. We are not aware of any specific climatic interpretation.

**Minor suggestions:**

**11   Lines 43-44: Thompson and Solomon 2002 do not show analyses of the Southern Annular Mode link to SWWA rainfall. This reference should be changed to Thompson et al. 2011:**

**115** **Thompson, D. W. J., S. Solomon, P. J. Kushner, M. H. England, K. M. Grise and D. J. Karoly, 2011: Signatures of the Antarctic ozone hole in Southern Hemisphere surface climate change, Nature Geoscience, 4, 741-749.**

Thank you for pointing this out. We apologize for this mistake. We have changed the reference accordingly at Lines 43-44: "...is a large-scale mode of climate variability that is correlated with rainfall in WA (Gong and Wang, 1999; Thompson et al., 2011; Fierro and Leslie, 2013)."

**120** **12  Line 58: Add a citation to Goodwin et al. 2004 here alongside the reference to van Ommen and Morgan 2010:**

**Goodwin, I. D., , T. D. van Ommen, , M. A. J. Curran, , and P. A. Mayewski, 2004: Mid latitude winter climate variability in the South Indian and southwest Pacific regions since 1300 ad. Climate Dyn., 22, 783–794.**

We have added this citation to the text at Line 58: "A relationship between rainfall in SWWA and the snowfall recorded in
**125** Dome Summit South (DSS) ice core drilling site on Law Dome, East Antarctica was found by Goodwin et al. (2004); van Ommen and Morgan (2010)."

**13  Lines 104-105: England et al. (2006) also analyse the quality of the Mk3L model simulations in relation to observed interannual variability of SWWA rainfall. Perhaps cite their findings here.**

Thank you. We have included these references, making the following change at Lines 104–105: "Both CSIRO Mk3 and CSIRO
**130** Mk3L produce credible simulations of large-scale precipitation, including over Australia (Cai et al., 2003; Cai and Shi, 2005; England et al., 2006; Phipps et al., 2011)."

**14  I found the shorthand term "MASK" a little obtuse for the region of statistical significance of the rainfall signal over SWWA. I think it is okay to identify that region upfront as the authors have done, but then just say that hereafter, when describing SWWA rainfall, it is taken to indicate the region delineated by this area in Fig. 1. I**
**135** **think the resulting text will be clearer that way.**

Thank you for this comment. We agree. We have made the following changes at Lines 164-166: "In order to quantify the MASK correlation coefficient and evaluate the statistical significance, we multiply the mask matrix (in the region has a value of 1 and outside the region has a value of 0) of the MASK with the AWAP gridded data to generate the MASK rainfall. Hereafter, the SWWA rainfall we are describing, is the MASK rainfall where the MASK region is delineated in Figure 1. Then
**140** we calculate the  SWWA rainfall correlation coefficient with DSS and test its statistical significance."We have changed all the "MASK rainfall" to "SWWA rainfall" after Line 166.

**15** **The amount of explained variance R2 should be added to Table 3, expressed as a percentage. I think this would add to the information provided in that table.**

Thank you for this comment. We agree. We have made the following changes to Table 3:

**Table 1.** The Pearson correlation coefficients for the  SWWA rainfall and the four BoM stations rainfall with the DSS snow accumulation. $R^2$ is the square of the correlation coefficient. All the correlations are statistically significant (6-year window, $p < 0.05$).

| Sample | Correlation coefficient | $R^2$ | Year (CE) |
|---|---|---|---|
|  SWWA | -0.597 | 0.356 | 1900–2015 |
| Arthur River | -0.548 | 0.300 | 1891–2015 |
| Boyanup | -0.623 | 0.388 | 1898–2013 |
| Cranbrook | -0.540 | 0.292 | 1891–2015 |
| Wonnaminta | -0.546 | 0.298 | 1905–2015 |

145 **16** **Lines 169-170: are there also non-significant stations within this MASK region? If so, the authors should point this out.**

Thank you for this comment. There are five non-significant stations within this MASK region (Figure 2). We have revised the manuscript accordingly at Lines 170–171: "... four stations are all geographically located in the MASK region (Figure 1) showing the consistency with the significance of MASK. Five other stations within the region have correlations of a similar
150 magnitude, but these correlations are not significant at the 5% probability level."

**17** **Typo, Figure 2 caption, line 1: plor -> plot**

Thank you for pointing this out. We have made the following change to the caption of Figure 2: "(a) The scatter  plot for AWAP rainfall in MASK region and DSS snow accumulation (both 6-year window) of period 1900 CE to 2015 CE with their linear model and 95% CI. (b) The histogram for model residuals using probability density function scaling. (c) The scatter plot
155 for model residuals and fitted data with their linear fit and 95% CI."

**18** **The term "model outputs" is used often in the paper, I would suggest changing this to terms like "the model simulations", "the model runs", "the model experiments" (etc.) throughout the manuscript.**

Thank you for this comment. We agree using the term "the model simulations" is better than "model outputs". We have changed all the "output/ outputs" to "simulation/ simulations" throughout the manuscript.

[Figure]

**Figure 2.** The correlation map for the southwest part of WA region for 6-year window AWAP rainfall and DSS snow accumulation from 1900 CE to 2015 CE. The outline area (green line) is the MASK region where the correlation is statistically significant (p < 0.05). Red diagonal line connecting 115°E 30°S and 120°E 35°S is the boundary of SWWA (van Ommen and Morgan, 2010). Perth is the capital city of WA. Boyanup, Wonnaminta, Arthur River and Cranbrook are the four significant (6-year window, p < 0.05) stations. Red "+" marked stations are the other 11 non-significant stations.

**19** **Figure E1: I understand why the authors may wish to include this diagram for completeness, but each of the panels looks basically the same as each of the other panels. So I wonder if there is any utility in this diagram being included?**

Thank you for this comment. We do not fully agree. Figure E1 supports the statement in the manuscript at Lines 285-288: "The simulated rainfall for each individual ensemble member is shown in Figure E1, with the corresponding CUSUM time series shown in Figure E2. Within each forced ensemble, there are considerable differences between the CUSUM time series for individual ensemble members. This highlights the role of unforced internal variability in driving SWWA rainfall, consistent with the findings of Cai and Shi (2005)." However, in regarding to this comment and also in regard to a comment by Referee 1, we will move all of the appendices to a supplementary document.

**References**

170    Cai, W. and Shi, G.: Multidecadal fluctuations of winter rainfall over southwest Western Australia simulated in the CSIRO Mark 3 coupled model, Geophysical Research Letters, 32, L12 701, https://doi.org/10.1029/2005GL022712, 2005.

Cai, W., Collier, M. A., Gordon, H. B., and Waterman, L. J.: Strong ENSO Variability and a Super-ENSO Pair in the CSIRO Mark 3 Coupled Climate Model, Monthly Weather Review, 131, 1189–1210, https://doi.org/10.1175/1520-0493(2003)131<1189:SEVAAS>2.0.CO;2, 2003.

175    England, M. H., Ummenhofer, C. C., and Santoso, A.: Interannual rainfall extremes over southwest Western Australia linked to Indian Ocean climate variability, Journal of Climate, 19, 1948–1969, 2006.

Fierro, A. O. and Leslie, L. M.: Links between central west Western Australian rainfall variability and large-scale climate drivers, Journal of climate, 26, 2222–2246, 2013.

Gong, D. and Wang, S.: Definition of Antarctic oscillation index, Geophysical research letters, 26, 459–462, 1999.

180    Goodwin, I., Van Ommen, T., Curran, M., and Mayewski, P.: Mid latitude winter climate variability in the South Indian and southwest Pacific regions since 1300 AD, Climate Dynamics, 22, 783–794, 2004.

Phipps, S., Rotstayn, L., Gordon, H., Roberts, J., Hirst, A., and Budd, W.: The CSIRO Mk 3 L climate system model version 1. 0-Part 1: Description and evaluation, Geoscientific Model Development, 4, 483–509, 2011.

Smith, I., McIntosh, P., Ansell, T., Reason, C., and McInnes, K.: Southwest Western Australian winter rainfall and its association with Indian Ocean climate variability, International Journal of Climatology: A Journal of the Royal Meteorological Society, 20, 1913–1930, 2000.

185

Thompson, D. W., Solomon, S., Kushner, P. J., England, M. H., Grise, K. M., and Karoly, D. J.: Signatures of the Antarctic ozone hole in Southern Hemisphere surface climate change, Nature geoscience, 4, 741–749, 2011.

Ummenhofer, C. C., Sen Gupta, A., Pook, M. J., and England, M. H.: Anomalous rainfall over southwest Western Australia forced by Indian Ocean sea surface temperatures, Journal of Climate, 21, 5113–5134, 2008.

190    van Ommen, T. D. and Morgan, V.: Snowfall increase in coastal East Antarctica linked with southwest Western Australian drought, Nature Geoscience, 3, 267–272, 2010.

---

## Author Response (AR1)

We thank the Editor and reviewers for their comments on the manuscript.

We have addressed the Editor's and reviewers' comments on the revised manuscript and marked up the changes on a track-changes version.

Kind regards,

Yaowen Zheng and the co-authors

---

## Referee Report (RR1)

Review of the revised paper:
"Extending and understanding the southwest Western Australian rainfall record using a snowfall reconstruction from Law Dome, East Antarctica"
By: Zheng, Jong, Phipps, Roberts, Moy, Curran, and van Ommen
Submitted to *Climate of the Past*

I have read the revised manuscript and also checked the authors' responses to my suggestions in the first review. I'm happy to recommend the revised manuscript for publication in CoP, but I include below a few final suggestions for edits before going to print. Line numbers here refer to the track-changes version of the latest revised manuscript.

1.   Line 52: Thank you for adding the discussion of other drivers of SWWA rainfall here — however the lead into this paragraph could be improved to be clearer on this point. Best to start with something along the lines:

"Variability in the SAM is not the sole driver of SWWA rainfall anomalies; for example local Indian Ocean sea surface temperatures can also play a role. In particular, negative sea surface temperature anomalies....."

2.   Table 3 and lines 202–209: Thank you for adding the explained variance % metrics in both the table and the text here. I agree that 30-40% is a sizeable contribution for reconstructing SWWA rainfall variability. The wording here could be tightened up a little though:

"The square of the correlation coefficients have shown the explained variance is maximum at around 30–40%. This is a significant fraction of the variance, although the tropics and subtropics can also play an important role in driving rainfall changes in SWWA (Smith et al., 2000; England et al., 2006; Ummenhofer et al., 2008). Thus using the Law Dome ice core snow accumulation proxy to reconstruct the SWWA rainfall focuses in on the SAM-related component of the rain-bearing systems, not the tropical / subtropical components. However, explaining 30–40% of SWWA rainfall variations is a valuable contribution to our ability to reconstruct past climate, so we construct a linear model for SWWA rainfall and DSS snow accumulation".

3.   Line 322 and Figure S7.  Thank you for following up on my suggestion to explore the latest BoM rainfall data during 2016 – 2020. Best to tighten the wording to read:

"This drought continued during 2015-2020 (Figure E7)".
because as time goes on, the drought may eventually be broken, and then the statement would become invalid.

4.  Nice final sentence added to the paper, thank you for sorting that.   This paper will make a valuable contribution to the literature.

Matthew England
UNSW

---------- END OF REVIEW ----------

---

## Author Response (AR2)

**Extending and understanding the South West Western Australian rainfall record using a snowfall reconstruction from Law Dome, East Antarctica**

by Yaowen Zheng, Lenneke M. Jong, Steven J. Phipps, Jason L. Roberts, Andrew D. Moy, Mark A. J. Curran, and Tas D. van Ommen,

submitted to Climate of the Past (https://doi.org/10.5194/cp-2020-124)

We thank the Editor and the Reviewer for the time they have spent reading and reviewing it. We respond to each of the suggestions for revision below. The reviewer's suggestions are shown in **bold text**, replies are shown in normal text, text from the original manuscript is shown in green, revisions of the Major Revision are shown in red, and proposed changes for the reconsideration to the manuscript are shown in blue.
* * *
**Editor's comment:**

**When I looked at your track changed version, it seemed that the scale of changes you had made was rather small and not really commensurate with what the two reviewers requested (both asked for major revisions). Reviewer 1 has looked at the paper again and confirms exactly this opinion. I noted in my previous editorial comment that your response did not answer the referee's request (point 3) about stationarity, and asked you to address this - but you did not. In view of this please prepare a further version of the manuscript, paying particular attention to the three points that the reviewer has emphasised. Once these are addressed, I will consider it worthwhile to ask the reviewers to spend time on the paper again. Thank you for acting on this.**

We thank the editor for these comments. We acknowledge that we could have explored the role of stationarity in further depth. In response to the comments by the editor and the reviewer, we have made further revisions to the manuscript. Most significantly, we have extended our statistical analysis of the robustness of the reconstruction and we have used the climate model simulations to explore the stability of the teleconnection between Law Dome and South West Western Australian precipitation. We offer a point-by-point response to the reviewers' comments below and attach a revised version of the manuscript.

**Reviewer's comments:**

**1** **For the mechanism that could explain the link between precipitation in SWWA and snowfall at Law Dome (point 1 in the comments), the authors mention the 'variations in meridional circulation south of Australia' without additional explanation. I could live with that as they refer to a previous paper but this is still very short. I am surprised that such a role of meridional circulation is not clearly mentioned in the introduction when discussing the processes influencing precipitation in SWWA or at Law Dome. By contrast, if I interpret well the figure 1 in the response (Figure S5) the 'Correlation between precipitation at Law Dome and Southern Hemisphere mean sea level pressure' displays a zonal structure with clear change in zonal winds but the meridional variations are not obvious to me. This is consistent with the sentence stating that 'the strength and position of southern hemisphere westerlies dominate changes in coastal Antarctica snowfall.' I am thus wondering if the justification implies a meridional change in zonal flow rather than the meridional circulation itself and more information on this would be very useful for the reader.**

We thank the reviewer for this comment. We have added more sentences in the Introduction starting at Line 80 to provide more information on the mechanism.

Lines 80–94: "at the DSS site (66.7697°S,112.8069°E,1370m elevation) (Roberts et al., 2015). Connections between mid-latitude climate and that of coastal East Antarctica have been reported for some time (Goodwin et al., 2004; van Ommen and Morgan, 2010). The strength and position of the southern hemisphere westerlies are important for coastal Antarctic snowfall rates, especially for cyclonically driven locations such as Law Dome, East Antarctica (Bromwich, 1988). The specific relationship between SWWA and snowfall recorded in the Dome Summit South (DSS) ice core from Law Dome was reported by van Ommen and Morgan (2010) who found a statistically significant anticorrelation between winter (JJA) mean SWWA regional rainfall and Law Dome snow accumulation. This link, which accounts for up to 40% of the shared variance on interannual to decadal timescales is associated with simultaneous anomalous northward flow of relatively cool, dry air to SWWA and southward flow of relatively warm, dry air to the Law Dome region. This teleconnection pattern is characterised by a broadly zonal wave three pattern in 500 hPa geopotential field. The northward flow to SWWA, while bringing showers to the southern coast, is distinct from the higher rainfall patterns which bring prefrontal rain from north/north-westerly directions (van Ommen and Morgan, 2010). More recently, this work has been extended using a longer 2035-year accumulation record from the Law Dome core (Roberts et al., 2015). As with the earlier accumulation record, this was dated by determining annual layers in the seasonally varying water stable isotope ratios and trace ions from multiple ice cores drilled at the DSS site (66.7697°S, 112.8069°E, 1370 m elevation) (Roberts et al., 2015). Taken together, the relationship between SWWA rainfall and DSS snowfall and the"

**2** **For the point two in my initial comments, the information provided in the response is interesting and the results**
50      **very convincily prove that the validation is fine if one considers the whole period. It is still important for me to**
     **compute the value of the correlation over the period 1900-1971 to see the effect of the trend at the end of the**
     **record. I am also still interested in a visual comparison of the time series of precipitation over SWWA and snow**
     **accumulation at Law Dome. One option would be to add the reconstruction (which is proportional to snow**
     **accumulation at Law Dome if I am right) on Figure S7 or alternatively the time series of observed rainfall on**
55      **Figure 3 (but this may be less clear on this scale).**

We thank the reviewer for this comment. We calculated the correlation over the period 1900–1970 and constructed the linear model. To present these results in context, we extended the analysis to construct a full jackknife ensemble in which all possible pairs of non-overlapping 45-year periods are omitted. This analysis has been incorporated into Table S4.

Although we find that the correlation between Law Dome accumulation and SWWA rainfall is weaker when the period
60 1971–2015 is omitted, we note that the 95% confidence intervals in the model parameters overlap between all members of the jackknife ensemble. At the 5% probability level, we cannot therefore reject the null hypothesis that the model parameters are unchanged when the period 1971–2015 is omitted. We conclude that there is no evidence to suggest that our model, and therefore our reconstruction, is biased by any post-1970 trend. We have added a corresponding paragraph in supplementary material Lines 54–58: "To further explore whether the relationship between Law Dome accumulation and SWWA rainfall has
65 changed during the current drought, the middle section of Table S4 repeats the jackknife analysis but increases the duration of the period omitted to 45 years. Although we find greater scatter in the values for the correlation coefficient, gradient and intercept, the 95% confidence intervals continue to overlap. We therefore find no evidence to suggest that the relationship between Law Dome accumulation and SWWA rainfall has changed.".

We have added an additional panel on Figure 2 for a visual comparison of the time series of precipitation over SWWA and
70 snow accumulation at Law Dome. We have also added a corresponding sentence in Lines 210–211: " The time series and the scatter plot for SWWA rainfall and DSS snow accumulation are shown in Figure 2a and Figure 2b, respectively. The data show a generally".

**3** **Still on the stationarity (point 3 in the initial comments), I am wondering why the authors do not provide, at least**
     **in their response, a running correlation between precipitation in SWWA and snowfall at Law Dome in the**
75      **simulations. It is a simple diagnostics. If the value remains high during the whole period, then it is a clear**
     **indication that the link is robust and stationary. If it changes a lot between centuries, at least a cautionary note**
     **should be added in the discussion.**

We thank the reviewer for this comment. We are sorry that we didn't clearly present our response on the running correlation in either the manuscript or the supplementary material. We added Figure S6 in the supplementary material and corresponding
80 sentences in Lines 64–70: "We further explored the link using the SWWA precipitation in the model simulations. To do this, we extract nine CSIRO Mk3L computing cells of the precipitation simulations around SWWA region and nine cells around

. **Table S4.** The correlation coefficient between the smoothed ice core record and the mean of the AWAP rainfall in the "MASK" region. The three sections represents the results for a 12–year jackknife, a 45–year jackknife and the full period, respectively. x1 and Intercept are the coefficients for the linear model: Rain = Snow * ( x1 ± 95% CI ) + Intercept ± 95% CI mm/year. The 95% confidence interval (CI) is estimated by multiplying the standard error of the model by 1.96.

| Section | Period | Correlation coefficient | x1 | Intercept (mm/year) |
|---|---|---|---|---|
| 12-year jackknife analysis | 1912–2015 | -0.536 | $-0.385 \pm 0.118$ | $667 \pm 85$ |
| | 1900–1911 & 1924–2015 | -0.513 | $-0.363 \pm 0.118$ | $651 \pm 85$ |
| | 1900–1923 & 1936–2015 | -0.532 | $-0.378 \pm 0.117$ | $662 \pm 85$ |
| | 1900–1935 & 1948–2015 | -0.524 | $-0.405 \pm 0.128$ | $684 \pm 93$ |
| | 1900–1947 & 1960–2015 | -0.545 | $-0.407 \pm 0.122$ | $685 \pm 88$ |
| | 1900–1959 & 1972–2015 | -0.499 | $-0.385 \pm 0.130$ | $671 \pm 94$ |
| | 1900–1971 & 1984–2015 | -0.506 | $-0.427 \pm 0.141$ | $699 \pm 101$ |
| | 1900–1983 & 1996–2015 | -0.492 | $-0.374 \pm 0.129$ | $662 \pm 92$ |
| | 1900–1995 & 2008–2015 | -0.512 | $-0.356 \pm 0.116$ | $651 \pm 82$ |
| | 1900–2007 | -0.593 | $-0.379 \pm 0.098$ | $670 \pm 71$ |
| 45-year jackknife analysis | 1945–2015 | -0.511 | $-0.351 \pm 0.139$ | $635 \pm 103$ |
| | 1900–1944 & 1990–2015 | -0.414 | $-0.389 \pm 0.202$ | $672 \pm 144$ |
| | 1900–1925 & 1971–2015 | -0.489 | $-0.403 \pm 0.170$ | $679 \pm 125$ |
| | 1900–1970 | -0.296 | $-0.193 \pm 0.147$ | $548 \pm 102$ |
| Full period | 1900–2015 | -0.529 | $-0.389 \pm 0.114$ | $672 \pm 82$ |

Law Dome region. We calculate the mean of each region's nine cells for each member of ensemble and each ensemble mean and perform 6-year smoothing for each series (for consistency), and then calculate the 100-year running correlation (Figure S6). This does not show a strong or sustained correlation. However, this is not surprising, given the mechanism outlined in van Ommen and Morgan (2010). Correlation is essentially connected with periods of enhanced meridional flow which will be less apparent over extended periods in which mix meridional and zonal modes of circulation."

However, we are aware that our last major revision was not clear enough to address this comment. We here add sentences in the manuscript corresponding to the supplementary material Section 5.

Lines 220–225:"and trend, and is generally symmetric along y = 0. We furthermore assess the robustness of the model in Supplement Section 5. We calculate the autocorrelation length (where the autocorrelation coefficient is equal or below zero) of the SWWA rainfall for the period 1900–2015 CE, and perform a jackknife (modified) analysis. The correlation coefficients of each individual jackknife ensemble member are all consistent with the period 1900–2015 CE (Supplement Table S4), and the 95% confidence intervals for the gradient and intercept are overlapped (Supplement Table S4), showing the statistically

[Figure]

**. Figure 2.** (a) The time series of anomalies for AWAP rainfall in the MASK region and DSS snow accumulation (6-year running averages) for the period 1900–2015 CE. (b) The scatter plot for AWAP rainfall in the MASK region and DSS snow accumulation (6-year running averages) for the period 1900–2015 CE with their linear model and 95% confidence interval. (c) The probability density function histogram for the model residuals. (d) The scatter plot for model residuals and fitted data with their linear fit and 95% confidence interval.

indistinguishable between each individual jackknife ensemble member. Taken together, there is no obvious evidence to reject

95    the linear model. Thus, we use".

Lines 265–273:"  We further validate the rainfall reconstruction in Supplement Section 5. We have found that the temporal stability of large-scale circulation over mid-latitude Australasian and higher latitude Law Dome regions is consistent, indicated by the correlation fields between Law Dome precipitation and mean sea level pressure in each member of the CSIRO Mk3L OGSV ensemble (Supplement Figure S5). This rainfall reconstruction has shown that

100    it is possible to use ice cores to investigate the longer-term context of rainfall variability over SWWA prior to the instrumental

era. We should also be aware that the Law Dome—SWWA precipitation correlation is stronger during periods of enhanced (stronger/more frequent) meridional circulation, but weaker during periods of weaker meridional circulation (van Ommen and Morgan, 2010). The robustness of the reconstruction may therefore be reduced during periods of reduced meridional flow between southern Australia and Law Dome (Supplement Section 5). We choose 1850 CE to be the year that separates before and after the"

[Figure]

**. Figure S6.** 100-year running correlation of the CSIRO Mk3L precipitation simulations between SWWA and Law Dome for each member and the each ensemble mean from 501 to 2000.

**References**

Bromwich, D. H.: Snowfall in high southern latitudes, Reviews of Geophysics, 26, 149–168, 1988.

Goodwin, I., Van Ommen, T., Curran, M., and Mayewski, P.: Mid latitude winter climate variability in the South Indian and southwest Pacific regions since 1300 AD, Climate Dynamics, 22, 783–794, 2004.

110   Roberts, J., Plummer, C., Vance, T., van Ommen, T., Moy, A., Poynter, S., Treverrow, A., Curran, M., and George, S.: A 2000-year annual record of snow accumulation rates for Law Dome, East Antarctica, Climate of the Past, 11, 697–707, 2015.

van Ommen, T. D. and Morgan, V.: Snowfall increase in coastal East Antarctica linked with southwest Western Australian drought, Nature Geoscience, 3, 267–272, 2010.

---

## Author Response (AR3)

**Extending and understanding the South West Western Australian rainfall record using a snowfall reconstruction from Law Dome, East Antarctica**

by Yaowen Zheng, Lenneke M. Jong, Steven J. Phipps, Jason L. Roberts, Andrew D. Moy, Mark A. J. Curran, and Tas D. van Ommen,

submitted to Climate of the Past (https://doi.org/10.5194/cp-2020-124)

We thank the Editor and the Reviewers for the time they have spent reading and reviewing it. They have improved this manuscript. We respond to each of the suggestions for revision below. The Editor/ Reviewers' suggestions are shown in **bold text**, replies are shown in normal text, text from the original manuscript is shown in blue and proposed changes to the manuscript are shown in purple.
* * *
**Editor's comment:**

**Dear authors**

**Thank you for the revisions you have made to your paper. As you will see on most issues the two reviewers are happy with the new version and believe it should be published. However I concur with rev 1's concern about the stationarity of the relationship between LD accumulation and precipitation in SWWA (as shown in Fig S6). In the supplement you dismiss this as hardly relevant, but it is precisely the relationship you assume in extending your reconstruction, and it deserves a fair comment in both the main text and the supplement.**

**Specifically rev 1 and I both understand that the instrumental data you have showed a good relationship over the last century between LD accumulation and precipitation in SWWA. In extending the reconstruction you assume that this remains the case, and maybe it does. However your climate model output in Fig S6 quote clearly shows that, at least in the model, the relationship can disappear or even show an opposite sign for century long periods. I appreciate that this doesn't affect your broader discussion about influences on SWWA rainfall, and that it could be an artefact in the model. But Fig S6 cannot just be dismissed: I am quite sure if the result had been the opposite that you would have made a big deal out of it.**

**I am therefore minded to accept the next version of your paper subject to:**

**a) you deal with the minor points raised by both reviewers and**

**25** **b) you add a statement of caution in the main text that the model does not show a stationary relationship (Fig S6) between LD accumulation and precipitation in SWWA, and that if this is really true it undermines the case for using LD accumulation alone as a proxy for SWWA rainfall. This should be borne in mind when assessing the reconstruction shown.**

**I would find it hard to accept a version that does not include such a statement in the main text, although of course you**
**30** **are welcome to argue against it if you feel you have a case.**

We thank the editor for these comments. We have respond to each of the minor points point-by-point below and attached a revised version of the manuscript. We agree that the statement of caution should be indicated in the manuscript.

To address this, we have added corresponding sentences in the manuscript Lines 255–257: " Dome (Supplement Section 5). Furthermore, the CSIRO Mk3L simulations exhibit variability in the correlation between Law Dome accumulation and SWWA
**35** precipitation on decadal to centennial timescales (Supplement Figure S6). Some caution is therefore needed in interpreting the reconstruction. We choose 1850 CE to be the year that separates before and after the", and in the supplementary material Lines 63–67: "We calculate the mean of each region's nine cells for each member of ensemble and each ensemble mean and perform 6-year smoothing for each series (for consistency), and then calculate the 100-year running correlation (Figure S6). The model simulations exhibit a lack of stationarity on decadal to centennial timescales, which should be considered when interpreting the
**40** reconstruction. The lack of a strong or consistent correlation between the simulated Law Dome and SWWA precipitation is not surprising, given the mechanism outlined in van Ommen and Morgan (2010). Correlation is essentially connected with"

**Reviewer 1's comments:**

**1  I would like to thank the reviewers for their detailed answers and additional information that allow a better interpretation of their results. If I consider than the answers to my first two points are satisfactory, I still have concerns about the point 3.**

45

**The authors show that the link between Law Dome accumulation and SWWA precipitation is stable and stationary over the past century (in particular in in the answer to my second point and lines 89-105 of the answer to point3). However, this is not the case for the climate model results over the past millennium. In the supplementary material lines 61-62, it is stated from climate model results that 'These show that the large scale circulation both matches the**

50

**pattern identified in van Ommen and Morgan (2010) and is stable through time'. We may agree (or not) that the large-scale simulated spatial pattern is stable in the examples shown Figure S5. It is clear that it is a prerequisite to use Law Dome accumulation as a robust proxy for SWWA precipitation (line 63 of the supplementary material). However, this is clearly not enough. The reconstruction is based on the link between Law Dome accumulation and precipitation in SWWA (line 197 of the main manuscript for instance), not between Law Dome accumulation and any large-scale**

55

**pattern. The correlation between the two variables (Law Dome and SWA precipitation) in the simulations varies between roughly -0.5 and +0.5 depending of the 100-yr period selected, showing a very strong non-stationarity in the link (Figure S6). The authors argue that it is not surprising but I wonder then how to justify using climate model results that Law Dome record can be used to reconstruct precipitation in SWWA for the whole period while there is no guarantee that the link will be the same in two different centuries. For instance, if the linear statistical model were**

60

**built from the climate model results using different 100-yr period instead of observations, we would obtain very different reconstructions.**

**I may admit that demonstrating the robustness of the link using modern data for the 20th century can be a justification for the statistical model but, as I understand it, the climate model results do not confirm at all the stationarity of this link for the whole millennium. The climate model may be wrong but at least it must be stated and**

65

**explained clearly. I thus strongly recommend adding a few lines stating that at the end of section 5 of the supplementary material. This conclusion on the instationarity if we consider climate model results, should also come as a caution on the robustness of the reconstruction in the main text as it is important for all the readers that may not check the supplementary information.**

We thank the reviewer for this comment. We have added the statement of caution in both main text and the supplementary

70   material. Please refer to our responses to the Editor's comment above.

**2  Line 75. Something seems missing before 'More recently, this work has been extended' as the previous lines discuss the teleconnection pattern while the sentence is devoted to the ice coring.**

We thank the reviewer for this comment. "This work" refers to the work has been done by van Ommen and Morgan (2010), which proposed a 750-year (1250–2000) DSS snow accumulation reconstruction. Roberts et al. (2015) extended this work and proposed a 2035-year (22BCE–2012CE) DSS snow accumulation record. To be more clear, we change the corresponding sentence in Lines 76–77: "rain from north/north-westerly directions (van Ommen and Morgan, 2010). More recently, van Ommen and Morgan (2010)'s work has been extended using a longer 2035-year accumulation record from the Law Dome core (Roberts et al., 2015). As"

**Reviewer 2's comments:**

**3  Line 52: Thank you for adding the discussion of other drivers of SWWA rainfall here—however the lead into this paragraph could be improved to be clearer on this point. Best to start with something along the lines: "Variability in the SAM is not the sole driver of SWWA rainfall anomalies; for example local Indian Ocean sea surface temperatures can also play a role. In particular, negative sea surface temperature anomalies....."**

Thank you for this comment. We have changed it accordingly.

Lines 50–51:"Variability in the SAM is not the sole driver of SWWA rainfall anomalies; for example local Indian Ocean sea surface temperatures can also play a role. In particular, negative sea surface temperature anomalies in the eastern Indian Ocean and positive"

**4  Table 3 and lines 202–209: Thank you for adding the explained variance % metrics in both the table and the text here. I agree that 30-40% is a sizeable contribution for reconstructing SWWA rainfall variability. The wording here could be tightened up a little though: "The square of the correlation coefficients have shown the explained variance is maximum at around 30–40%. This is a significant fraction of the variance, although the tropics and subtropics can also play an important role in driving rainfall changes in SWWA (Smith et al., 2000; England et al., 2006; Ummenhofer et al., 2008). Thus using the Law Dome ice core snow accumulation proxy to reconstruct the SWWA rainfall focuses in on the SAM-related component of the rain-bearing systems, not the tropical / subtropical components. However, explaining 30–40% of SWWA rainfall variations is a valuable contribution to our ability to reconstruct past climate, so we construct a linear model for SWWA rainfall and DSS snow accumulation".**

Thank you for this comment. We have changed it accordingly.

Lines 186–191:"at the 5% probability level (Supplement Figure S4). The squares of the correlation coefficients show that the explained variance is around 30-40%. We note that the tropics and subtropics can also play an important role in driving rainfall changes in SWWA (Smith et al., 2000; England et al., 2006; Ummenhofer et al., 2008). Using Law Dome accumulation to reconstruct SWWA rainfall therefore focuses on the SAM-related component of the rain-bearing systems, not the tropical or subtropical components. However, our analysis shows that it makes a valuable contribution to our ability to reconstruct past climate and so we construct a linear model for SWWA rainfall and DSS snow accumulation."

**5  Line 322 and Figure S7. Thank you for following up on my suggestion to explore the latest BoM rainfall data during 2016 – 2020. Best to tighten the wording to read:**

**"This drought continued during 2015-2020 (Figure E7)".**

**because as time goes on, the drought may eventually be broken, and then the statement would become invalid.**

Thank you for this comment. We agree. We have changed it accordingly.

Lines 306–307:"reduction in the mean rainfall at around 1971 CE resulted in a prolonged drought in SWWA. This drought continued during 2015–2020 (Supplement Figure S7). To highlight dry epochs of an equivalent duration to the observed drought to date, we

**6  Nice final sentence added to the paper, thank you for sorting that. This paper will make a valuable contribution to the literature.**

Thank you very much. Your comments have improved this manuscript.

**References**

Roberts, J., Plummer, C., Vance, T., van Ommen, T., Moy, A., Poynter, S., Treverrow, A., Curran, M., and George, S.: A 2000-year annual
record of snow accumulation rates for Law Dome, East Antarctica, Climate of the Past, 11, 697–707, 2015.

van Ommen, T. D. and Morgan, V.: Snowfall increase in coastal East Antarctica linked with southwest Western Australian drought, Nature
Geoscience, 3, 267–272, 2010.

120